# DISTRIBUTIONAL GENERALIZATION: STRUCTURE BEYOND TEST ERROR

## ABSTRACT

Classifiers in machine learning are often reduced to single dimensional quantities, such as test error or loss. Here, we initiate a much richer study of classifiers by considering the entire joint distribution of their inputs and outputs. We present both new empirical behaviors of standard classifiers, as well as quantitative conjectures which capture these behaviors. Informally, our conjecture states: the output distribution of an interpolating classifier matches the distribution of true labels, when conditioned on certain subgroups of the input space. For example, if we mislabel 30% of dogs as cats in the train set of CIFAR-10, then a ResNet trained to interpolation will in fact mislabel roughly 30% of dogs as cats on the *test set* as well, while leaving other classes unaffected. This conjecture has implications for the theory of overparameterization, scaling limits, implicit bias, and statistical consistency. Further, it can be seen as a new kind of generalization, which goes beyond measuring single-dimensional quantities to measuring entire distributions.

## 1 INTRODUCTION

In learning theory, when we study how well a classifier "generalizes", we usually consider a single metric – its test error (Shalev-Shwartz & Ben-David, 2014). However, there could be many different classifiers with the same test error that differ substantially in, say, the subgroups of inputs on which they make errors. Reducing classifiers to a single number misses these rich aspects of their behavior. In this work, we propose formally studying the entire *joint distribution* of classifier inputs and outputs. That is, the distribution $(x, f(x))$ for samples from the distribution $x \sim D$ for a classifier $f(x)$. This distribution reveals many structural properties of the classifier beyond test error (such as *where* the errors occur). In fact, we discover new behaviors of modern classifiers that can only be understood in this framework. As an example, consider the following experiment (Figure 1).

**Experiment 1.** *Consider a binary classification version of CIFAR-10, where CIFAR-10 images $x$ have binary labels* `Animal/Object`. *Take 50K samples from this distribution as a train set, but apply the following label noise: flip the label of cats to* `Object` *with probability 30%. Now train a WideResNet $f$ to 0 train error on this train set. How does the trained classifier behave on test samples? Options below:*

**(1)** The test error is low across all classes, since there is only 3% overall label noise in the train set.

**(2)** Test error is "spread" across the animal class. After all, the classifier is not explicitly told what a cat or a dog is, just that they are all animals.

**(3)** The classifier misclassifies roughly 30% of test cats as "objects", but all other animals are largely unaffected.

The reality is closest to option (3) as shown in Figure 1. The left panel shows the joint density of train inputs $x$ with train labels `Object/Animal`. The right panel shows the *classifier predictions* $f(x)$ on *test inputs $x$*.

There are several notable things about this experiment. First, the error is *localized* to cats in the test set as it was in the train set, even though no explicit cat labels were provided. The interpolating model is thus sensitive to subgroup-structures in the distribution. Second, the *amount* of error on the cat class is close to the noise applied on the train set. Thus, the behavior of the classifier on the train set *generalizes* to the test set in a stronger sense than just average error. Specifically, when

Figure 1: The setup and result of Experiment 1. The CIFAR-10 train set is labeled as either Animals or Objects, with label noise affecting only cats. A WideResNet-28-10 is then trained to 0 train error on this train set, and evaluated on the test set. Full experimental details in Appendix C.2

*conditioned on a subgroup* (cat), the *distribution* of the true labels is close to that of the classifier outputs. Third, this is not the behavior of the Bayes-optimal classifier, which would always output the maximum-likelihood label instead of reproducing the noise in the distribution. The network is thus behaving poorly from the perspective of Bayes-optimality, but behaving well in a certain distributional sense (which we will formalize soon).

Now, consider a seemingly unrelated experimental observation. Take an AlexNet trained on ImageNet, a 1000-way classification problem with 116 varieties of dogs. AlexNet only achieves 56.5% test accuracy on ImageNet. However, it at least classifies most dogs as *some* variety of dog (with 98.4% accuracy), though it may mistake the exact breed. In this work, we show that both of these experiments are examples of the same underlying phenomenon. We empirically show that for an interpolating classifier, its classification outputs are close in distribution to the true labels — even when conditioned on many subsets of the domain. For example, in Figure 1, the distribution of $p(f(x)|x = \text{cat})$ is close to the true label distribution of $p(y|x = \text{cat})$. We propose a formal conjecture (Feature Calibration), that predicts which subgroups of the domain can be conditioned on for the above distributional closeness to hold.

These experimental behaviors could not have been captured solely by looking at average test error, as is done in the classical theory of generalization. In fact, they are special cases of a new kind of generalization, which we call "Distributional Generalization".

## 1.1 DISTRIBUTIONAL GENERALIZATION

Informally, Distributional Generalization states that the outputs of classifiers $f$ on their train sets and test sets are close *as distributions* (as opposed to close in just error). That is, the following joint distributions[1] are close:

$$(x, f(x))_{x \sim \text{TestSet}} \approx (x, f(x))_{x \sim \text{TrainSet}} \tag{1}$$

The remainder of this paper is devoted to making the above statement precise, and empirically checking its validity on real-world tasks. Specifically, we want to formally define the notion of approximation ($\approx$), and understand how it depends on the problem parameters (the type of classifier, number of train samples, etc). We focus primarily on interpolating methods, where we formalize Equation (1) through our Feature Calibration Conjecture.

## 1.2 OUR CONTRIBUTIONS AND ORGANIZATION

We discover new empirical behaviors of interpolating classifiers, and we propose quantitative conjectures to characterize these behaviors.

- In Section 3, we introduce a "Feature Calibration" conjecture, which unifies our experimental observations. Roughly, Feature Calibration says that the outputs of classifiers match the statistics of their training distribution when conditioned on certain subgroups.

- In Section 4, we experimentally stress test our Feature Calibration conjecture across various settings in machine learning, including neural networks, kernel machines, and decision trees. This highlights the universality of our results across machine learning.

---

[1]These distributions also include the randomness in sampling the train and test sets, and in training the classifier, as we define more precisely in Section 3.

- In Section 5, we relate our results to classical generalization, by defining a new notion of Distributional Generalization which extends and subsumes the classical notion.

- Finally, in Section 5.2 we informally discuss how Distributional Generalization can be applied even for non-interpolating methods.

Our results extend our scientific understanding of of interpolating methods, and introduce a new type of generalization exhibited across many methods in machine learning.

## 1.3 Related Work and Significance

Our work has connections to, and implications for many existing research programs in deep learning.

**Implicit Bias and Overparameterization.** There has been a long line of recent work towards understanding overparameterized and interpolating methods, since these pose challenges for classical theories of generalization (e.g. Zhang et al. (2016); Belkin et al. (2018a;b; 2019); Liang & Rakhlin (2018); Nakkiran et al. (2020); Schapire et al. (1998); Breiman (1995); Soudry et al. (2018); Gunasekar et al. (2018)). The "implicit bias" program here aims to answer: *Among all models with 0 train error, which model is actually produced by SGD?* Most existing work seeks to characterize the exact implicit bias of models under certain (sometimes strong) assumptions on the model, training method or the data distribution. In contrast, our conjecture applies across many different interpolating models (from neural nets to decision trees) as they would be used in practice, and thus form a sort of "universal implicit bias" of these methods. Moreover, our results place constraints on potential future theories of implicit bias, and guide us towards theories that better capture practice.

**Benign Overfitting.** Most prior works on interpolating classifiers attempt to explain why training to interpolation "does not harm" the the model. This has been dubbed "benign overfitting" (Bartlett et al., 2020) and "harmless interpolation" (Muthukumar et al., 2020), reflecting the widely-held belief that interpolation does not harm the decision boundary of classifiers. In contrast, we find that interpolation actually does "harm" classifiers, in predictable ways: fitting the label noise on the train set causes similar noise to be reproduced at test time. Our results thus indicate that interpolation can significantly affect the decision boundary of classifiers, and should not be considered a purely "benign" effect.

**Classical Generalization and Scaling Limits.** Our framework of Distributional Generalization is insightful even to study classical generalization, since it reveals much more about models than just their test error. For example, statistical learning theory attempts to understand if and when models will asymptotically converge to Bayes optimal classifiers, in the limit of large data ("asymptotic consistency" Shalev-Shwartz & Ben-David (2014); Wasserman (2013)). In deep learning, there are at least two distinct ways to scale model and data to infinity together: the *underparameterized* scaling limit, where data-size $\gg$ model-size always, and the *overparameterized* scaling limit, where data-size $\ll$ model-size always. The underparameterized scaling limit is well-understood: when data is essentially infinite, neural networks will converge to the Bayes-optimal classifier (provided the model-size is large enough, and the optimization is run for long enough, with enough noise to escape local minima). On the other hand, our work suggests that in the *overparameterized* scaling limit, models will *not* converge to the Bayes-optimal classifier. Specifically, our Feature Calibration Conjecture implies that in the limit of large data, interpolating models will approach a *sampler* from the distribution. That is, the limiting model $f$ will be such that the output $f(x)$ is a sample from $p(y|x)$, as opposed to the Bayes-optimal $f^*(x) = \mathrm{argmax}_y\, p(y|x)$. This claim— that overparameterized models do not converge to Bayes-optimal classifiers— is unique to our work as far as we know, and highlights the broad implications of our results.

**Locality and Manifold Learning.** Our intuition for the behaviors in this work is that they arise due to some form of "locality" of the trained classifiers, in an appropriate embedding space. For example, the behavior observed in Experiment 1 would be consistent with that of a 1-Nearest-Neighbor classifier in a embedding that separates the CIFAR-10 classes well. This intuition that classifiers learn good embeddings is present in various forms in the literature, for example: the so-called called "manifold hypothesis," that natural data lie on a low-dimensional manifold (Narayanan & Mitter, 2010; Sharma & Kaplan, 2020), as well as works on local stiffness of the loss landscape (Fort et al., 2019), and works showing that overparameterized neural networks can learn hidden low-dimensional structure

in high-dimensional settings (Gerace et al., 2020; Bach, 2017; Chizat & Bach, 2020). It is open to more formally understand connections between our work and the above.

**Uncertainty Estimation.** Since the appearance of the current work on arXiv, it has been directly built on by other authors. The work of Jiang et al. (2021) investigates our conjectures further, and extends them to develop a method for out-of-distribution uncertainty estimation. This highlights the fundamental nature and importance of our results, since they have already been used in a practical application. A full discussion of related works is in Appendix A.

## 2 PRELIMINARIES

**Notation.** We consider joint distributions $\mathcal{D}$ on $x \in \mathcal{X}$ and discrete $y \in \mathcal{Y} = [k]$. Let $S = \{(x_i, y_i)\}_{i=1}^n \sim \mathcal{D}^n$ denote a train set of $n$ iid samples from $\mathcal{D}$. Let $\mathcal{A}$ denote the training procedure (including architecture and training algorithm for neural networks), and let $f \leftarrow \text{Train}_{\mathcal{A}}(S)$ denote training a classifier $f$ on train-set $S$ using procedure $\mathcal{A}$. We consider classifiers which output hard decisions $f : \mathcal{X} \rightarrow \mathcal{Y}$. Let $\text{NN}_S(x) = x_i$ denote the nearest-neighbor to $x$ in train-set $S$, with respect to a distance metric $d$. Our theorems will apply to any distance metric, and so we leave this unspecified. Let $\text{NN}_S^{(y)}(x)$ denote the nearest-neighbor estimator itself, that is, $\text{NN}_S^{(y)}(x) := y_i$ where $x_i = \text{NN}_S(x)$.

**Experimental Setup.** Briefly, we train all classifiers to interpolation (to 0 train error). Neural networks (MLPs and ResNets (He et al., 2016)) are trained with SGD. Interpolating decision trees are trained using the growth rule from Random Forests (Breiman, 2001). For kernel classification, we consider kernel regression on one-hot labels and kernel SVM, with small or 0 of regularization (which is often optimal (Shankar et al., 2020)). Full experimental details are provided in Appendix B.

**Distributional Closeness.** We formalize distributional closeness using the notion of Integral Probability Metrics (Müller, 1997), which we review here. For two distributions $P, Q$ over $\mathcal{X} \times \mathcal{Y}$, let a "test" (or "distinguisher") be a function $T : \mathcal{X} \times \mathcal{Y} \rightarrow [0, 1]$ which accepts a sample from either distribution, and is intended to classify the sample as either from distribution $P$ or $Q$. For any set $\mathcal{C} \subseteq \{T : \mathcal{X} \times \mathcal{Y} \rightarrow [0, 1]\}$ of tests, we say distributions $P$ and $Q$ are "$\varepsilon$-indistinguishable up to $\mathcal{C}$-tests" if they are close with respect to all tests in class $\mathcal{C}$. That is,

$$P \approx_\varepsilon^{\mathcal{C}} Q \iff \sup_{T \in \mathcal{C}} \left| \mathbb{E}_{(x,y) \sim P}[T(x,y)] - \mathbb{E}_{(x,y) \sim Q}[T(x,y)] \right| \leq \varepsilon \qquad (2)$$

Total-Variation distance is equivalent to closeness in all tests, i.e. $\mathcal{C} = \{T : \mathcal{X} \times \mathcal{Y} \rightarrow [0, 1]\}$, but we consider closeness for restricted families of tests $\mathcal{C}$. $P \approx_\varepsilon Q$ denotes $\varepsilon$-closeness in TV-distance.

## 3 FEATURE CALIBRATION CONJECTURE

### 3.1 DISTRIBUTIONS OF INTEREST

We first define three key distributions that we will use in stating our formal conjecture. For a given data distribution $\mathcal{D}$ over $\mathcal{X} \times \mathcal{Y}$ and training procedure $\text{Train}_{\mathcal{A}}$, we consider the following three distributions over $\mathcal{X} \times \mathcal{Y}$:

1. **Source $\mathcal{D}$:** $(x, y)$ where $x, y \sim \mathcal{D}$.
2. **Train $\mathcal{D}_{\text{tr}}$:** $(x_{\text{tr}}, f(x_{\text{tr}}))$ where $S \sim \mathcal{D}^n, f \leftarrow \text{Train}_{\mathcal{A}}(S), (x_{\text{tr}}, y_{\text{tr}}) \sim S$
3. **Test $\mathcal{D}_{\text{te}}$:** $(x, f(x))$ where $S \sim \mathcal{D}^n, f \leftarrow \text{Train}_{\mathcal{A}}(S), x, y \sim \mathcal{D}$

The source distribution $\mathcal{D}$ is simply the original distribution. To sample once from the **Train Distribution $\mathcal{D}_{\text{tr}}$**, we first sample a train set $S \sim \mathcal{D}^n$, train a classifier $f$ on it, then output $(x_{\text{tr}}, f(x_{\text{tr}}))$ for a random *train point* $x_{\text{tr}} \in S$. That is, $\mathcal{D}_{\text{tr}}$ is the distribution of input and outputs of a trained classifier $f$ on its train set. To sample once from the **Test Distribution $\mathcal{D}_{\text{te}}$**, we do this same procedure, but output $(x, f(x))$ for a random *test point* $x$. That is, the $\mathcal{D}_{\text{te}}$ is the distribution of input and outputs of a trained classifier $f$ at test time. The only difference between the Train Distribution and

Test Distribution is that the point $x$ is sampled from the train set or the test set, respectively.[2] For interpolating classifiers, $f(x_{\mathrm{tr}}) = y_{\mathrm{tr}}$ on the train set, and so the Source and Train distributions are equivalent: $\mathcal{D} \equiv \mathcal{D}_{\mathrm{tr}}$. (Note that these definitions, crucially, involve randomness from sampling the train set, training the classifier, and sampling a test point).

## 3.2 FEATURE CALIBRATION

We now formally describe the Feature Calibration Conjecture. At a high level, we argue that the distributions $\mathcal{D}_{\mathrm{te}}$ and $\mathcal{D}$ are statistically close for interpolating classifiers if we first "coarsen" the domain of $x$ by some partition $L : \mathcal{X} \to [M]$ in to $M$ parts. That is, for certain partitions $L$, the following distributions are statistically close:

$$(L(x), f(x))_{x \sim \mathcal{D}} \approx_{\varepsilon} (L(x), y)_{x \sim \mathcal{D}}$$

We think of $L$ as defining subgroups over the domain— for example, $L(x) \in \{\text{dog, cat, horse}\dots\}$. Then, the above statistical closeness is essentially equivalent to requiring that for all subgroups $\ell \in [M]$, the conditional distribution of classifier output on the subgroup—$p(f(x)|L(x) = \ell)$ — is close to the true conditional distribution: $p(y|L(x) = \ell)$.

The crux of our conjecture lies in defining exactly which subgroups $L$ satisfy this distributional closeness, and quantifying the $\varepsilon$ approximation. This is subtle, since it must depend on almost all parameters of the problem. For example, consider a modification to Experiment 1, where we use a fully-connected network (MLP) instead of a ResNet. An MLP cannot properly distinguish cats even when it is actually provided the real CIFAR-10 labels, and so (informally) it has no hope of behaving differently on cats in the setting of Experiment 1, where the cats are not labeled explicitly (See Figure C.2 for results with MLPs). Similarly, if we train the ResNet with very few samples from the distribution, the network will be unable to recognize cats. Thus, the allowable partitions must depend on the classifier family and the training method, including the number of samples.

We conjecture that allowable partitions are those which can themselves be learnt to good test performance with an identical training procedure, but trained with the labels of the partition $L$ instead of $y$. To formalize this, we define a *distinguishable feature*: a partition of the domain $\mathcal{X}$ that is learnable for a given training procedure. Thus, in Experiment 1, the partition into CIFAR-10 classes would be a distinguishable feature for ResNets (trained with SGD with 50K or more samples), but not for MLPs. The definition below depends on the training procedure $\mathcal{A}$, the data distribution $\mathcal{D}$, number of train samples $n$, and an approximation parameter $\varepsilon$ (which we think of as $\varepsilon \approx 0$).

**Definition 1** (($\varepsilon, \mathcal{A}, \mathcal{D}, n$)-Distinguishable Feature). *For a distribution $\mathcal{D}$ over $\mathcal{X} \times \mathcal{Y}$, number of samples $n$, training procedure $\mathcal{A}$, and small $\varepsilon \geq 0$, an $(\varepsilon, \mathcal{A}, \mathcal{D}, n)$-distinguishable feature is a partition $L : \mathcal{X} \to [M]$ of the domain $\mathcal{X}$ into $M$ parts, such that training a model using $\mathcal{A}$ on $n$ samples labeled by $L$ works to classify $L$ with high test accuracy. Precisely, $L$ is a $(\varepsilon, \mathcal{A}, \mathcal{D}, n)$-distinguishable feature if:*

$$\Pr_{\substack{S = \{(x_i, L(x_i)\}_{x_1, \dots, x_n \sim \mathcal{D}} \\ f \leftarrow \mathrm{Train}_{\mathcal{A}}(S); \ x \sim \mathcal{D}}} [f(x) = L(x)] \geq 1 - \varepsilon$$

This definition depends only on the marginal distribution of $\mathcal{D}$ on $x$, and not on the label distribution $p_{\mathcal{D}}(y|x)$. To recap, this definition is meant to capture a labeling of the domain $\mathcal{X}$ that is learnable for a given training procedure $\mathcal{A}$. It must depend on the architecture used by $\mathcal{A}$ and number of samples $n$, since more powerful classifiers can distinguish more features. Note that there could be many distinguishable features for a given setting $(\varepsilon, \mathcal{A}, \mathcal{D}, n)$ — including features not implied by the class label such as the presence of grass in a CIFAR-10 image. Our main conjecture follows.

**Conjecture 1** (Feature Calibration). *For all natural distributions $\mathcal{D}$, number of samples $n$, interpolating training procedures $\mathcal{A}$, and $\varepsilon \geq 0$, the following distributions are statistically close for all $(\varepsilon, \mathcal{A}, \mathcal{D}, n)$-distinguishable features $L$:*

$$\underset{\substack{f \leftarrow \mathrm{Train}_{\mathcal{A}}(\mathcal{D}^n); \ x, y \sim \mathcal{D}}}{(L(x), f(x))} \quad \approx_{\varepsilon} \quad \underset{x, y \sim \mathcal{D}}{(L(x), y)} \tag{3}$$

---

[2] Technically, these definitions require training a fresh classifier for each sample, using independent train sets. For practical reasons most of our experiments train a single classifier $f$ and evaluate it on the entire train/test set.

*or equivalently:*

$$(L(x), \widehat{y})_{x, \widehat{y} \sim \mathcal{D}_{\text{te}}} \quad \approx_\varepsilon \quad (L(x), y)_{x, y \sim \mathcal{D}} \tag{4}$$

This claims that the TV distance between the LHS and RHS of Equation (4) is at most $\varepsilon$, where $\varepsilon$ is the error of the distinguishable feature (in Definition 1). We claim that this holds *for all* distinguishable features $L$ "automatically" – we simply train a classifier, without specifying any particular partition. The formal statements of Definition 1 and Conjecture 1 may seem somewhat arbitrary, involving many quantifiers over $(\varepsilon, \mathcal{A}, \mathcal{D}, n)$. However, we believe these statements are natural: In addition to extensive experimental evidence in Section 4, we also prove that Conjecture 1 is formally true as stated for 1-Nearest-Neighbor classifiers in Theorem 1.

### 3.3 FEATURE CALIBRATION FOR 1-NEAREST-NEIGHBORS

Here we prove that the 1-Nearest-Neighbor classifier formally satisfies Conjecture 1, under mild assumptions. We view this theorem as support for our (somewhat involved) formalism of Conjecture 1. Indeed, without Theorem 1 below, it is unclear if our statement of Conjecture 1 can ever be satisfied by any classifier, or if it is simply too strong to be true. This theorem applies generically to a wide class of distributions; the only assumption is a weak regularity condition. The proof of Theorem 1 is straightforward, and provided in Appendix D – but this strong property of nearest-neighbors was not know before, to our knowledge.

**Theorem 1.** *Let $\mathcal{D}$ be a distribution over $\mathcal{X} \times \mathcal{Y}$, and let $n \in \mathbb{N}$ be the number of train samples. Assume the following regularity condition holds: Sampling the nearest-neighbor train point to a random test point yields (close to) a uniformly random test point. That is, suppose that for some small $\delta \geq 0$, the distributions: $\{\text{NN}_S(x)\}_{\substack{S \sim \mathcal{D}^n \\ x \sim \mathcal{D}}} \quad \approx_\delta \quad \{x\}_{x \sim \mathcal{D}}$. Then, Conjecture 1 holds. That is, for all $(\varepsilon, \text{NN}, \mathcal{D}, n)$-distinguishable partitions $L$, the following distributions are statistically close:*

$$\{(y, L(x))\}_{x, y \sim \mathcal{D}} \quad \approx_{\varepsilon + \delta} \quad \{(\text{NN}_S^{(y)}(x), L(x)\}_{\substack{S \sim \mathcal{D}^n \\ x, y \sim \mathcal{D}}} \tag{5}$$

### 3.4 LIMITATIONS: NATURAL DISTRIBUTIONS

Technically, Conjecture 1 is not fully specified, since it does not specify exactly which classifiers or distributions obey the conjecture. We do not claim that *all* classifiers and distributions satisfy our conjectures. Nevertheless, we claim our conjectures hold in all "natural" settings, which informally means settings with real data and classifiers that are actually used in practice. The problem of understanding what separates "natural distributions" from artificial ones is not unique to our work, and lies at the heart of deep learning theory. Many theoretical works handle this by considering simplified distributional assumptions (e.g. smoothness, well-separatedness, gaussianity), which are mathematically tractable, but unrealistic in practice (Arora et al., 2019; Li et al., 2019; Allen-Zhu et al., 2018). In contrast, we do not make unrealistic mathematical assumptions. This benefit of realism comes at the cost of mathematical formalism. We hope that as the theory of deep learning evolves, we will better understand how to formalize the notion of "natural" in our conjectures.

## 4 EXPERIMENTS: FEATURE CALIBRATION

We now give empirical evidence for our conjecture in a variety of settings in machine learning. In each experiment, we consider a feature that is (verifiably) distinguishable, and then test our Feature Calibration conjecture for this feature. Each of the experimental settings below highlights a different aspect of interpolating classifiers, which may be of independent interest. Selected experiments are summarized here, with full details and further experiments in Appendix C.

**Constant Partition:** Consider the trivially-distinguishable *constant* feature: $L(x) = 0$ everywhere. For this feature, Conjecture 1 reduces to the statement that the marginal distribution of class labels for any interpolating classifier is close to the true marginals $p(y)$. To test this, we construct a variant of CIFAR-10 with class-imbalance and train classifiers with varying levels of test errors to interpolation on it. As shown in Figure 2B, the marginals of the classifier outputs are close to the true marginals, even for a classifier that only achieves 37% test error.

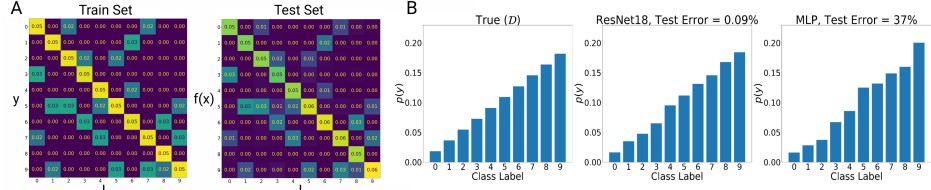

Figure 2: **Feature Calibration. (A)** Random confusion matrix on CIFAR-10, with a WideResNet28-10 trained to interpolation. Left: Joint density of labels $y$ and original class $L$ on the train set. Right: Joint density of classifier predictions $f(x)$ and original class $L$ on the test set. These two joint densities are close, as predicted by Conjecture 1. **(B)** Constant partition: The CIFAR-10 train set is class-rebalanced according to the left panel distribution. The center and right panels show that both ResNets and MLPs have the correct marginal distribution of outputs, even though the MLP has high test error.

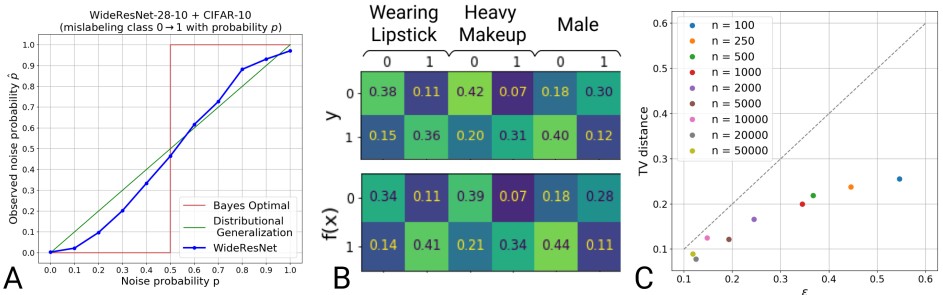

Figure 3: **Feature Calibration. (A)** CIFAR-10 with $p$ fraction of class $0 \to 1$ mislabeled on the train set. Plotting observed noise on classifier outputs vs. applied noise on the train set. **(B)** Multiple feature calibration on CelebA. **(C)** TV-distance between $(L(x), f(x))$ and $(L(x), y)$ for a variant of Experiment 1 with error on the distinguishable partitions ($\varepsilon$). The error was changed by changing the number of samples $n$.

**Coarse Partition:** Consider AlexNet trained on ILSVRC-2012 ImageNet (Russakovsky et al., 2015), a 1000-class image classification problem with 116 varieties of dogs. The network achieves only 56.5% accuracy on the test set. But it will at least classify most dogs as dogs (with 98.4% accuracy), making $L(x) \in \{$dog, not-dog$\}$ a distinguishable feature. Moreover, as predicted by Conjecture 1, the network is *calibrated* with respect to dogs: 22.4% of all dogs in ImageNet are Terriers, and indeed the network classifies 20.9% of all dogs as Terriers (though it has 9% error on which specific dogs it classifies as Terriers). See Appendix Table 2 for details, and related experiments on ResNets and kernels in Appendix C.

**Class Partition:** We now consider settings where the class labels are themselves distinguishable features (eg: CIFAR-10 classes are distinguishable by ResNets). Here our conjecture predicts the behavior of interpolating classifiers under structured label noise. As an example, we generate a random spare confusion matrix and apply this to the labels of CIFAR-10 as shown in Figure 2A. We find that a WideResNet trained to interpolation outputs the same confusion matrix on the test set as well (Figure 2B). Now, to test that this phenomenon is indeed robust to the level of noise, we mislabel class $0 \to 1$ with probability $p$ in the CIFAR-10 train set for varying levels of $p$. We then observe $\widehat{p}$, the fraction of samples mislabeled by this network from $0 \to 1$ in the test set (Figure 3A shows $p$ versus $\widehat{p}$). The Bayes optimal classifier for this distribution behaves as a step function (in red), and a classifier that obeys Conjecture 1 exactly would follow the diagonal (in green). The actual experiment (in blue) is close to the behavior predicted by Conjecture 1. This experiment shows a contrast with classical learning theory. While most existing theory focuses on whether classifiers converge to the Bayes optimal solution, we show that interpolating classifiers behave "optimally" in a different sense: they match the distribution of their train set. We discuss this further in Section 5. See Appendix C.4 for more experiments, including other classifiers such as Decisions Trees.

**Multiple features:** Conjecture 1 states that the network should be automatically calibrated for all distinguishable features, without any explicit labels for them. To do this, we use the CelebA dataset (Liu et al., 2015), containing images with many binary attributes per image. ("male", "blond hair", etc). We train a ResNet-50 to classify one of the hard attributes (accuracy 80%) and confirm that the Feature Calibration holds for all the other attributes (Figure 3) that are themselves distinguishable.

**Quantitative predictions:** We now test the quantitative predictions made by Conjecture 1. This conjecture states that the TV-distance between the joint distributions $(L(x), f(x))$ and $(L(x), y)$ is at most $\varepsilon$, where $\varepsilon$ is the error of the training procedure in learning $L$ (see Definition 1). To test this, we consider binary task similar to Experiment 1 where (Ship, Plane) are labeled as class 0 and (Cat, Dog) are labeled as class 1, with $p = 0.3$ fraction of cats mislabeled to class 0. Then, we train a network to interpolation on this task. To vary the error $\varepsilon$ systematically, we train networks with varying number of train samples. Networks with fewer samples have larger $\varepsilon$ since they are worse at classifying the distinguishable features of (Ship,Plane,Cat,Dog). Then, we use the same setup to train networks on the binary task and measure the TV-distance between $(L(x), f(x))$ and $(L(x), y)$ in this task. The results are shown in Figure 3C. As predicted, the TV distance on the binary task is upper bounded by $\varepsilon$ error on the 4-way classification task.

## 5 DISTRIBUTIONAL GENERALIZATION

In order to relate our results to the classical theory of generalization, we now propose a formal definition of "Distributional Generalization", which subsumes both Feature Calibration and classical generalization. A trained model $f$ obeys classical generalization (with respect to test error) if its error on the train set is close to its error on the test distribution. We first rewrite this using our definitions below.

**Classical Generalization (informal):** *Let $f$ be a trained classifier. Then $f$ generalizes if:*

$$\mathop{\mathbb{E}}_{\substack{x \sim TrainSet \\ \widehat{y} \leftarrow f(x)}} [\mathbb{1}\{\widehat{y} \neq y(x)\}] \approx \mathop{\mathbb{E}}_{\substack{x \sim TestSet \\ \widehat{y} \leftarrow f(x)}} [\mathbb{1}\{\widehat{y} \neq y(x)\}] \tag{6}$$

Above, $y(x)$ is the true class of $x$ and $\widehat{y}$ is the predicted class. The LHS of Equation 6 is the train error of $f$, and the RHS is the test error. Using our definitions of $\mathcal{D}_{\text{tr}}, \mathcal{D}_{\text{te}}$ from Section 3.1, and defining $T_{\text{err}}(x, \widehat{y}) := \mathbb{1}\{\widehat{y} \neq y(x)\}$, we can write Equation 6 equivalently:

$$\mathop{\mathbb{E}}_{x, \widehat{y} \sim \mathcal{D}_{\text{tr}}} [T_{\text{err}}(x, \widehat{y})] \approx \mathop{\mathbb{E}}_{x, \widehat{y} \sim \mathcal{D}_{\text{te}}} [T_{\text{err}}(x, \widehat{y})] \tag{7}$$

That is, classical generalization states that a certain function ($T_{\text{err}}$) has similar expectations on both the Train Distribution $\mathcal{D}_{\text{tr}}$ and Test Distribution $\mathcal{D}_{\text{te}}$. We can now introduce Distributional Generalization, which is a property of trained classifiers. It is parameterized by a set of bounded functions ("tests"): $\mathcal{T} \subseteq \{T : \mathcal{X} \times \mathcal{Y} \to [0, 1]\}$.

**Distributional Generalization:** *Let $f$ be a trained classifier. Then $f$ satisfies Distributional Generalization with respect to tests $\mathcal{T}$ if:*

$$\forall T \in \mathcal{T} : \quad \mathop{\mathbb{E}}_{x, \widehat{y} \sim \mathcal{D}_{\text{tr}}} [T(x, \widehat{y})] \approx \mathop{\mathbb{E}}_{x, \widehat{y} \sim \mathcal{D}_{\text{te}}} [T(x, \widehat{y})] \tag{8}$$

This states that the train and test distribution have similar expectations for *all* functions in the family $\mathcal{T}$, which we can write as: $\mathcal{D}_{\text{tr}} \approx^{\mathcal{T}} \mathcal{D}_{\text{te}}$. For the singleton set $\mathcal{T} = \{T_{\text{err}}\}$, this is equivalent to classical generalization, but it may hold for much larger sets $\mathcal{T}$. This definition of Distributional Generalization, like the definition of classical generalization, is just defining an object— not stating when or how it is satisfied. Feature Calibration turns this into a concrete conjecture.

### 5.1 FEATURE CALIBRATION AS DISTRIBUTIONAL GENERALIZATION

We can write our Feature Calibration Conjecture as a special case of Distributional Generalization, for a certain family of tests $\mathcal{T}$. Informally, for a given setting, the family $\mathcal{T}$ is all tests which take input $(x, y)$, but only depend on $x$ via a *distinguishable feature* (Definition 1). For example, a test of the form $T(x, y) = g(L(x), y)$ where $L$ is a distinguishable feature, and $g$ is arbitrary. Formally,

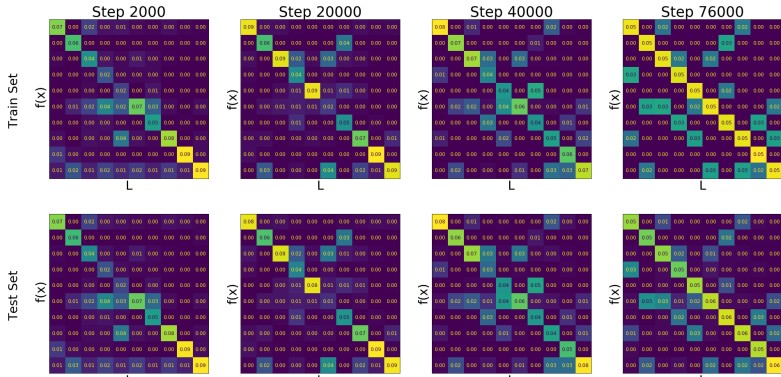

Figure 4: **Distributional Generalization for WideResNet on CIFAR-10.** The confusion matrices on the train set (top row) and test set (bottom row) remain close throughout training.

for a given problem setting, suppose $\mathcal{L}$ is the set of $(\varepsilon, \mathcal{A}, \mathcal{D}, n)$-distinguishable features. Then Conjecture 1 states that $\forall L \in \mathcal{L} : (L(x), f(x)) \approx_\varepsilon (L(x), y)$. This is equivalent to the statement

$$\mathcal{D}_{\text{te}} \approx_\varepsilon^{\mathcal{T}} \mathcal{D} \tag{9}$$

where $\mathcal{T}$ is the set of functions $\mathcal{T} := \{T : T(x, y) = g(L(x), y), \ L \in \mathcal{L}, \ g : \mathcal{X} \times \mathcal{Y} \to [0, 1]\}$. For interpolating classifiers, we have $\mathcal{D} \equiv \mathcal{D}_{\text{tr}}$, and so Equation (9) is equivalent to $\mathcal{D}_{\text{te}} \approx_\varepsilon^{\mathcal{T}} \mathcal{D}_{\text{tr}}$, which is a statement of Distributional Generalization. Since any classifier family will contain a large number of distinguishable features, the set $\mathcal{L}$ may be very large. Hence, the distributions $\mathcal{D}_{\text{tr}}$ and $\mathcal{D}_{\text{te}}$ can be thought of as being close *as distributions*.

## 5.2 BEYOND INTERPOLATING METHODS

The previous sections focused on *interpolating* classifiers, which fit their train sets exactly. Here we informally discuss how to extend our results beyond interpolating methods. The discussion in this section is not as precise as in previous sections, and is only meant to suggest that our abstraction of Distributional Generalization can be useful in other settings. For non-interpolating classifiers, we may still expect that they behave similarly on their test and train sets – that is, $\mathcal{D}_{\text{te}} \approx^{\mathcal{T}} \mathcal{D}_{\text{tr}}$ for some family of tests $\mathcal{T}$. For example, the following is a possible generalization of Feature Calibration.

**Conjecture 2** (Generalized Feature Calibration, informal)**.** *For trained classifiers $f$, the following distributions are statistically close for many partitions $L$ of the domain:*

$$\begin{array}{ccc} (L(x), \widehat{y}) & \approx & (L(x), \widehat{y}) \\ {\scriptstyle x, \widehat{y} \sim \mathcal{D}_{\text{te}}} & & {\scriptstyle x, \widehat{y} \sim \mathcal{D}_{\text{tr}}} \end{array} \tag{10}$$

We do not yet understand how to state this conjecture formally, but we give experimental evidence in its support. In Figure 4, we apply label noise from a random sparse confusion to the CIFAR-10 train set. We then train a single WideResNet28-10, and measure its predictions on the train and test over increasing SGD steps. The top row shows the confusion matrix of predictions $f(x)$ vs true labels $L(x)$ on the train set, and the bottom row shows the corresponding confusion matrix on the test set. As the network is trained for longer, it fits more of the noise on the train set, and this noise is mirrored almost identically on the test set. Full experimental details in Appendix B.

## 6 CONCLUSION

This work initiates the study of a new kind of generalization— Distributional Generalization— which considers the entire input-output behavior of classifiers, instead of just their test error. We presented both new empirical behaviors, and new formal conjectures which characterize these behaviors. Roughly, our conjecture states that the outputs of classifiers on the test set are "close in distribution" to their outputs on the train set. These results build a deeper understanding of models used in practice, and we hope our results inspire further work on distributional generalization in machine learning.

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

# A  FULL RELATED WORK

Our work is inspired by the broader study of interpolating and overparameterized methods in machine learning; a partial list of works in this theme includes Zhang et al. (2016); Belkin et al. (2018a;b; 2019); Liang & Rakhlin (2018); Nakkiran et al. (2020); Mei & Montanari (2019); Schapire et al. (1998); Breiman (1995); Ghorbani et al. (2019); Hastie et al. (2019); Bartlett et al. (2020); Advani & Saxe (2017); Geiger et al. (2019); Gerace et al. (2020); Chizat & Bach (2020); Goldt et al. (2019); Arora et al. (2019); Allen-Zhu et al. (2019); Neyshabur et al. (2018); Dziugaite & Roy (2017); Muthukumar et al. (2020); Neal et al. (2018).

**Interpolating Methods.** Many of the best-performing techniques on high-dimensional tasks are interpolating methods, which fit their train samples to 0 train error. This includes neural-networks and kernels on images (He et al., 2016; Shankar et al., 2020), and random forests on tabular data (Fernández-Delgado et al., 2014). Interpolating methods have been extensively studied both recently and in the past, since we do not theoretically understand their practical success (Schapire et al., 1998; Schapire, 1999; Breiman, 1995; Zhang et al., 2016; Belkin et al., 2018a;b; 2019; Liang & Rakhlin, 2018; Mei & Montanari, 2019; Hastie et al., 2019; Nakkiran et al., 2020). In particular, much of the classical work in statistical learning theory (uniform convergence, VC-dimension, Rademacher complexity, regularization, stability) fails to explain the success of interpolating methods (Zhang et al., 2016; Belkin et al., 2018a;b; Nagarajan & Kolter, 2019). The few techniques which do apply to interpolating methods (e.g. margin theory (Schapire et al., 1998)) remain vacuous on modern neural-networks and kernels.

**Decision Trees.** In a similar vein to our work, Wyner et al. (2017); Olson & Wyner (2018) investigate decision trees, and show that random forests are equivalent to a Nadaraya–Watson smoother Nadaraya (1964); Watson (1964) with a certain smoothing kernel. Decision trees (Breiman et al., 1984) are often intuitively thought of as "adaptive nearest-neighbors," since they are explicitly a spatial-partitioning method (Hastie et al., 2009). Thus, it may not be surprising that decision trees behave similarly to 1-Nearest-Neighbors. Wyner et al. (2017); Olson & Wyner (2018) took steps towards characterizing and understanding this behavior – in particular, Olson & Wyner (2018) defines an equivalent smoothing kernel corresponding to a random forest, and empirically investigates the quality of the conditional density estimate. Our work presents a formal characterization of the quality of this conditional density estimate (Conjecture 1), which is a novel characterization even for decision trees, as far as we know.

**Kernel Smoothing.** The term kernel regression is sometimes used in the literature to refer to kernel *smoothers*, such as the Nadaraya–Watson kernel smoother (Nadaraya, 1964; Watson, 1964). But in this work we use the term "kernel regression" to refer only to regression in a Reproducing Kernel Hilbert Space, as described in the experimental details.

**Label Noise.** Our conjectures also describe the behavior of neural networks under label noise, which has been empirically and theoretically studied in the past, though not formally characterized before (Zhang et al., 2016; Belkin et al., 2018b; Rolnick et al., 2017; Natarajan et al., 2013; Thulasidasan et al., 2019; Ziyin et al., 2020; Chatterji & Long, 2020). Prior works have noticed that vanilla interpolating networks are sensitive to label noise (e.g. Figure 1 in Zhang et al. (2016), and Belkin et al. (2018b)), and there are many works on making networks more robust to label noise via modifications to the training procedure or objective (Rolnick et al., 2017; Natarajan et al., 2013; Thulasidasan et al., 2019; Ziyin et al., 2020). In contrast, we claim this sensitivity to label noise is not necessarily a problem to be fixed, but rather a consequence of a stronger property: distributional generalization.

**Conditional Density Estimation.** Our density calibration property is similar to the guarantees of a conditional density estimator. More specifically, Conjecture 1 states that an interpolating classifier *samples* from a distribution approximating the conditional density of $p(y|x)$ in a certain sense. Conditional density estimation has been well-studied in classical nonparametric statistics (e.g. the Nadaraya–Watson kernel smoother (Nadaraya, 1964; Watson, 1964)). However, these classical methods behave poorly in high-dimensions, both in theory and in practice. There are some attempts to extend these classical methods to modern high-dimensional problems via augmenting estimators with neural networks (e.g. Rothfuss et al. (2019)). Random forests have also been known to exhibit properties similar to conditional density estimators. This has been formalized in various ways, often only with asymptotic guarantees (Meinshausen, 2006; Pospisil & Lee, 2018; Athey et al., 2019).

No prior work that we are aware of attempts to characterize the quality of the resulting density estimate via testable assumptions, as we do with our formulation of Conjecture 1. Finally, our motivation is not to design good conditional density estimators, but rather to study properties of interpolating classifiers — which we find happen to share properties of density estimators.

Feature Calibration (Conjecture 1) is also related to the concepts of calibration and multicalibration (Guo et al., 2017; Niculescu-Mizil & Caruana, 2005; Hébert-Johnson et al., 2018). In our framework, calibration is implied by Feature Calibration for a specific set of partitions $L$ (determined by level sets of the classifier's confidence). However, we are not concerned with a specific set of partitions (or "subgroups" in the algorithmic fairness literature) but we generally aim to characterize for which partitions Feature Calibration holds. Moreover, we consider only hard-classification decisions and not confidences, and we study only standard learning algorithms which are not given any distinguished set of subgroups/partitions in advance. Our notion of distributional generalization is also related to the notion of "distributional subgroup overfitting" introduced recently by Yaghini et al. (2019) to study algorithmic fairness. This can be seen as studying distributional generalization for a specific family of tests (determined by distinguished subgroups in the population).

**Locality and Manifold Learning.** Our intuition for the behaviors in this work is that they arise due to some form of "locality" of the trained classifiers, in an appropriate space. This intuition is present in various forms in the literature, for example: the so-called called "manifold hypothesis," that natural data lie on a low-dimensional manifold (e.g. Narayanan & Mitter (2010); Sharma & Kaplan (2020)), as well as works on local stiffness of the loss landscape (Fort et al., 2019), and works showing that overparameterized neural networks can learn hidden low-dimensional structure in high-dimensional settings (Gerace et al., 2020; Bach, 2017; Chizat & Bach, 2020). It is open to more formally understand connections between our work and the above.

**Note about Proper Scoring Rules:** If the loss function used in training is a *strictly-proper scoring rule* such as cross-entropy, then we may expect that in the limit of a large-capacity network and infinite data, training on samples $\{(x_i, y_i)\}$ will yield a good density estimate of $p(y|x)$ at the softmax layer. However, this is not what is happening in our experiments: First, our experiments consider the hard-decisions, not the softmax outputs. Second, we observe Conjecture 1 even in settings without proper scoring rules (kernel SVM and decision trees).

## B  EXPERIMENTAL DETAILS

Here we describe general background, and experimental details common to all sections. Then we provide section-specific details below.

### B.1  DATASETS

We consider the image datasets CIFAR-10 and CIFAR-100 (Krizhevsky et al., 2009), MNIST (LeCun et al., 1998), Fashion-MNIST (Xiao et al., 2017), CelebA (Liu et al., 2015), and ImageNet (Russakovsky et al., 2015). We normalize images to $x \in [0, 1]^{C \times W \times H}$.

We also consider tabular datasets from the UCI repository Dua & Graff (2017). For UCI data, we consider the 121 classification tasks as standardized in Fernández-Delgado et al. (2014). Some of these tasks have very few examples, so we restrict to the 92 classification tasks from Fernández-Delgado et al. (2014) which have at least 200 total examples.

### B.2  MODELS

We consider neural-networks, kernel methods, and decision trees.

#### B.2.1  DECISION TREES

We train interpolating decision trees using a growth rule from Random Forests (Breiman, 2001; Ho, 1995): selecting a split based on a random $\sqrt{d}$ subset of $d$ features, splitting based on Gini impurity, and growing trees until all leafs have a single sample. This is as implemented by Scikit-learn Pedregosa et al. (2011) defaults with `RandomForestClassifier` (`n_estimators=1, bootstrap=False`).

### B.2.2 KERNELS

Throughout this work we consider classification via kernel regression and kernel SVM. For $M$-class classification via kernel regression, we follow the methodology in e.g. Rahimi & Recht (2008); Belkin et al. (2018b); Shankar et al. (2020). We solve the following convex problem for training:

$$\alpha^* := \operatorname*{argmin}_{\alpha \in \mathbb{R}^{N \times M}} ||K\alpha - y||_2^2 + \lambda \alpha^T K \alpha$$

where $K_{ij} = k(x_i, x_j)$ is the kernel matrix of the training points for a kernel function $k$, $y \in \mathbb{R}^{N \times M}$ is the one-hot encoding of the train labels, and $\lambda \geq 0$ is the regularization parameter. The solution can be written

$$\alpha^* = (K + \lambda I)^{-1} y$$

which we solve numerically using SciPy `linalg.solve` (Virtanen et al., 2020). We use the explicit form of all kernels involved. That is, we do not use random-feature approximations (Rahimi & Recht, 2008), though we expect they would behave similarly.

The kernel predictions on test points are then given by

$$g_\alpha(x) := \sum_{i \in [N]} \alpha_i k(x_i, x) \tag{11}$$

$$f_\alpha(x) := \operatorname*{argmax}_{j \in [M]} g_\alpha(x)_j \tag{12}$$

where $g(x) \in \mathbb{R}^M$ are the kernel regressor outputs, and $g(x) \in [M]$ is the thresholded classification decision. This is equivalent to training $M$ separate binary regressors (one for each label), and taking the argmax for classification. We usually consider *unregularized* regression ($\lambda = 0$), except in Section 5.2.

For kernel SVM, we use the implementation provided by Scikit-learn (Pedregosa et al., 2011) `sklearn.svm.SVC` with a precomputed kernel, for inverse-regularization parameter $C \geq 0$ (larger $C$ corresponds to smaller regularization).

**Types of Kernels.** We use the following kernel functions $k : \mathbb{R}^d \times \mathbb{R}^d \to \mathbb{R}_{\geq 0}$.

- Gaussian Kernel (RBF): $k(x_i, x_j) = \exp(-\frac{||x_i - x_j||_2^2}{2\tilde{\sigma}^2})$.
- Laplace Kernel: $k(x_i, x_j) = \exp(-\frac{||x_i - x_j||_2}{\tilde{\sigma}})$.
- Myrtle10 Kernel: This is the compositional kernel introduced by Shankar et al. (2020). We use their exact kernel for CIFAR-10.

For the Gaussian and Laplace kernels, we parameterize bandwidth by $\sigma := \tilde{\sigma}/\sqrt{d}$. We use the following bandwidths, found by cross-validation to maximize the unregularized test accuracy:

- MNIST: $\sigma = 0.15$ for RBF kernel.
- Fashion-MNIST: $\sigma = 0.1$ for RBF kernel. $\sigma = 1.0$ for Laplace kernel.
- CIFAR-10: Myrtle10 Kernel from Shankar et al. (2020), and $\sigma = 0.1$ for RBF kernel.

### B.2.3 NEURAL NETWORKS

We use 4 different neural networks in our experiments. We use a multi-layer perceptron, and three different Residual networks.

**MLP:** We use a Multi-layer perceptron or a fully connected network with 3 hidden layers with 512 neurons in each layer. A hidden layer is followed by a BatchNormalization layer and ReLU activation function.

**WideResNet:** We use the standard WideResNet-28-10 described in Zagoruyko & Komodakis (2016). Our code is based on this repository.

**ResNet50:** We use a standard ResNet-50 from the PyTorch library (Paszke et al., 2017).

|  | MLP | ResNet18 | WideResNet | ResNet50 |
|---|---|---|---|---|
| **Batchsize** | 128 | 128 | 128 | 32 |
| **Epochs** | 820 | 200 | 200 | 50 |
| **Optimizer** | Adam $(\beta_1 = 0.9, \beta_2 = 0.999)$ | SGD + Momentum (0.9) | SGD + Momentum (0.9) | SGD |
| **Learning rate** (LR) schedule | Constant LR = 0.001 | Inital LR= 0.05 scale by 0.1 at epochs $(80, 120)$ | Inital LR= 0.1 scale by 0.2 at epochs $(80, 120, 160)$ | Initial LR = 0.001, scale by 0.1 if training loss stagnant for 2000 gradient steps |
| **Data Augmentation** | Random flips + RandomCrop(32, padding=4) | | | |
| **CIFAR-10 Error** | $\sim 37\%$ | $\sim 8\%$ | $\sim 4\%$ | N/A |

Table 1: Hyperparameters used to train the neural networks and their errors on the unmodified CIFAR-10 dataset

**ResNet18:** We use a modification of ResNet18 He et al. (2016) adapted to CIFAR-10 image sizes. Our code is based on this repository.

For Experiment 1 and Section 4, the hyperparameters used to train the above networks are given in Table 1.

# C    FEATURE CALIBRATION: APPENDIX

## C.1    A GUIDE TO READING THE PLOTS

All the experiments in support of Conjecture 1 involve various quantities which we enumaerate here

1. Inputs $x$: Each experiment involves inputs from a standard dataset like CIFAR-10 or MNIST. We use the standard train/test splits for every dataset.

2. Distinguishable feature $L(x)$: This feature depends only on input $x$. We consider various features like the original classes itself, a superset of classes (as in coarse partition) or some secondary attributes (like the binary attributes provided with CelebA)

3. Output labels $y$: The output label may be some modification of the original labels. For instance, by adding some type of label noise, or a constructed binary task as in Experiment 1

4. Classifier family $F$: We consider various types of classifiers like neural networks trained with gradient based methods, kernel and decision trees.

In each experiment, we are interested in two joint densities $(y, L(x))$, which depends on our dataset and task and is common across train and test, and $(f(x), L(x))$ which depends on the interpolating classifiers outputs on the *test* set. Since $y, L(x)$ and $f(x)$ are discrete, we will look at their discrete joint distributions. We sometimes refer to $(y, L(x))$ as the train joint density, as at interpolation $(y, L(x)) = (f(x), L(x))$ for all training inputs $x$. We also refer to $(f(x), L(x))$ as the test density, as we measure this only on the test set.

## C.2    EXPERIMENT 1

**Experimental details:** We now provide further details for Experiment 1. We first construct a dataset from CIFAR-10 that obeys the joint density $(y, L(x))$ shown in Figure 1 left panel. We then train a WideResNet-28-10 (WRN-28-10) on this modified dataset to zero training error. The network is trained with the hyperparameters described in Table 1. We then observe the joint density $(f(x), L(x))$ on the test images and find that the two joint densities are close as shown in Figure 5.

We now consider a modification of this experiment as follows:

**Experiment 2.** *Consider the following distribution over images $x$ and binary labels $y$. Sample $x$ as a uniformly random CIFAR-10 image, and sample the label as $p(y|x) = Bernoulli(\texttt{CIFAR\_Class(x)}/10)$. That is, if the CIFAR-10 class of $x$ is $k \in \{0, 1, \ldots 9\}$, then the label is 1 with probability $(k/10)$ and 0 otherwise. Figure 5 shows this joint distribution of $(x, y)$. As before, train a WideResNet to 0 training error on this distribution.*

In this experiment too, we observe that the train and test joint densities are close as shown in Figure 5.

Now, we repeat the same experiment, but with an MLP instead of WRN-28-10. The training procedure is described in Table 1. This MLP has an error on 37% on the original CIFAR-10 dataset.

Since this MLP has poor accuracy on the original CIFAR-10 classification task, it does not form a distinguishable partition for it. As a result, the train and test joint densities (Figure 6) do not match as well as they did for WRN-28-10.

## C.3    CONSTANT PARTITION

Conjecture 1 states that the marginal distribution of class labels for any interpolating classifier $f(x)$ is close to the true marginals $p(y)$. To show this, we construct a dataset based on CIFAR-10 that has class-imbalance. For class $k \in \{0...9\}$, sample $(k + 1) \times 500$ images from that class. This will give us a dataset where classes will have marginal distribution $p(y = \ell) \propto \ell + 1$ for classes $\ell \in [10]$, as shown in Figure 2. We do this both for the training set and the test set, to keep the distribution $\mathcal{D}$ fixed.

We then train a variety of classifiers (MLPs, RBF Kernel, ResNets) to interpolation on this dataset, which have varying levels of test errors (9-41%). The class balance of classifier outputs on the (rebalanced) test set

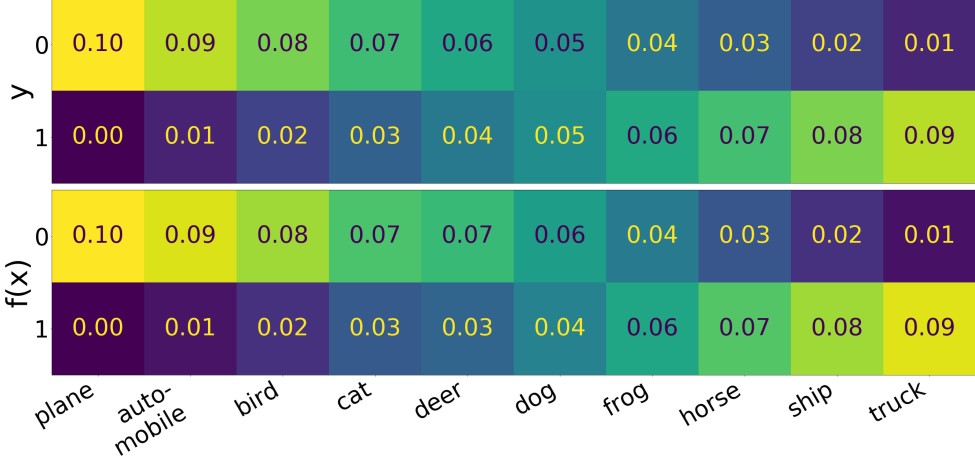

Figure 5: **Distributional Generalization in Experiment 2.** Joint densities of the distributions involved in Experiment 2. The top panel shows the joint density of labels on the train set: $(\texttt{CIFAR\_Class(x)}, y)$. The bottom panels shows the joint density of classifier predictions on the test set: $(\texttt{CIFAR\_Class(x)}, f(x))$. Distributional Generalization claims that these two joint densities are close.

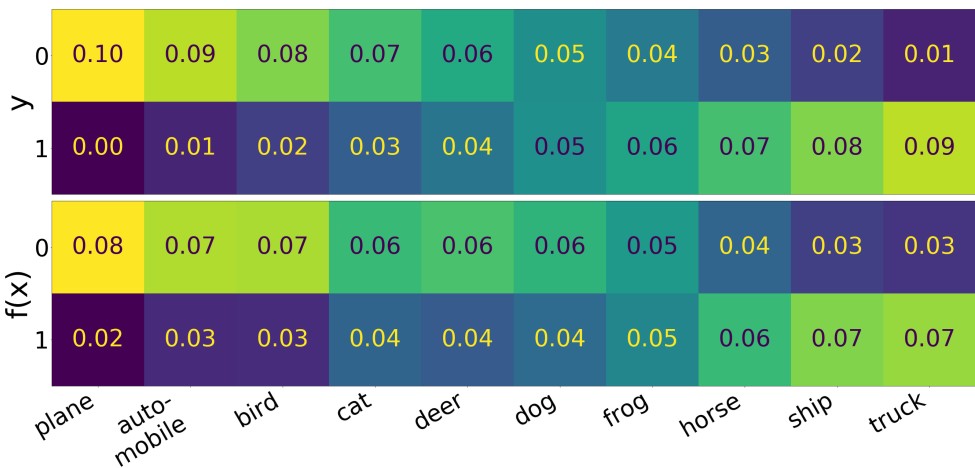

Figure 6: Joint density of $(y, \text{Class}(x))$, top, and $(f(x), \text{Class}(x))$, bottom, for test samples $(x, y)$ from Experiment 2 for an MLP.

### C.4 CLASS PARTITION

#### C.4.1 NEURAL NETWORKS AND CIFAR-10

We now describe details for the experiments in Figures 2A and 3A. A WRN-28-10 achieves an error of $4\%$ on CIFAR-10. Hence, the original labels in CIFAR-10 form a distinguishable partition for this dataset. To demonstrate that Conjecture 1 holds, we consider different structured label noise on the CIFAR-10 dataset. To do so, we apply a variety of confusion matrices to the data. That is, for a confusion matrix $C : 10 \times 10$ matrix, the element $c_{ij}$ gives the joint density that a randomly sampled image had original label $j$, but is flipped to class $i$. For no noise, this would be an identity matrix.

We begin by a simple confusion matrix where we flip only one class $0 \to 1$ with varying probability $p$. Figure 7A shows one such confusion matrix for $p = 0.4$. We then train a WideResNet-28-10 to zero train error on this dataset. We use the hyperparameters described in B.2 We find that the

classifier outputs on the test set closely track the confusion matrix that was applied to the distribution. Figure 7C shows that this is independent of the value of $p$ and continues to hold for $p = [0, 1]$.

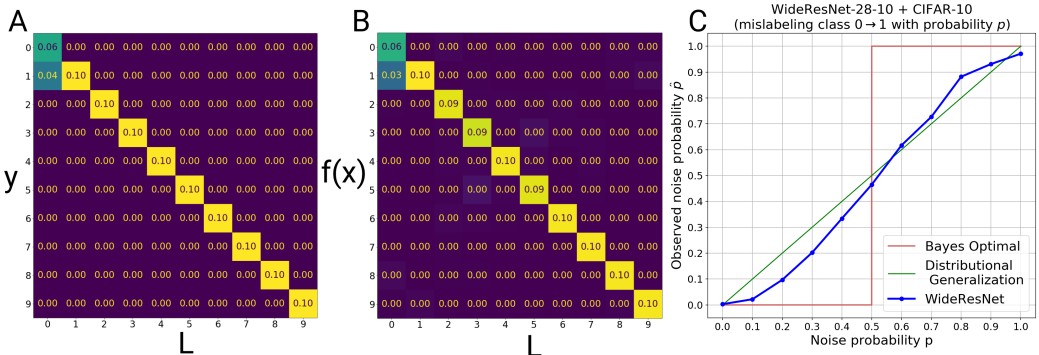

Figure 7: **Feature Calibration with original classes on CIFAR-10**: We train a WRN-28-10 on the CIFAR-10 dataset where we mislabel class $0 \to 1$ with probability $p$. (A): Joint density of the distinguishable features $L$ (the original CIFAR-10 class) and the classification task labels $y$ on the train set for noise probability $p = 0.4$. (B): Joint density of the original CIFAR-10 classes $L$ and the network outputs $f(x)$ on the test set. (C): Observed noise probability in the network outputs on the test set (the $(1, 0)$ entry of the matrix in B) for varying noise probabilities $p$

To show that this is not dependent on the particular class used, we also show that the same holds for a random confusion matrix. We generate a sparse confusion matrix as follows. We set the diagonal to $0.5$. Then, for every class $j$, we pick any two random classes for and set them to $0.2$ and $0.3$. We train a WRN-28-10 on it and report the test confusion matrix. The resulting train and test densities are shown in Figure 2A. As expected, the train and test confusion matrices are close, and share the same sparsity pattern.

### C.4.2 Decision Trees

Figure 8 shows a version of this experiment for decision trees on the molecular biology UCI task. The molecular biology task is a 3-way classification problem: to classify the type of a DNA splice junction (donor, acceptor, or neither), given the sequence of DNA (60 bases) surrounding the junction. We add varying amounts of label noise that flips class 2 to class 1 with a certain probability, and we observe that interpolating decision trees reproduce this same structured label noise on the test set.

Similar results hold for decision trees; here we show experiments on two UCI tasks: `wine` and `mushroom`.

The `wine` task is a 3-way classification problem: to identify the cultivar of a given wine (out of 3 cultivars), given 13 physical attributes describing the wine. Figure 9 shows an analogous experiment with label noise taking class 1 to class 2.

The `mushroom` task is a 2-way classification problem: to classify the type of edibility of a mushroom (edible vs poisonous) given 22 physical attributes (e.g. stalk color, odor, etc). Figure 10 shows an analogous experiment with label noise flipping class 0 to class 1.

### C.5 Multiple Features

Conjecture 1 states that the network should be automatically calibrated for all distinguishable features, without any explicit labels for them. To verify this, we use the CelebA dataset (Liu et al., 2015), containing images with various labelled binary attributes per-image ("male", "blond hair", etc). Some of these attributes form a distinguishable feature for ResNet50 as they are learnable to high accuracy (Jahandideh et al., 2018). We pick one of hard attributes as the target classification task. We train a ResNet-50 to predict the attribute {Attractive, Not Attractive}. We choose this attribute because a ResNet-50 performs poorly on this task (test error $\sim 20\%$) and has good class balance. We choose an attribute with poor generalization because the conjecture would hold trivially for if the network

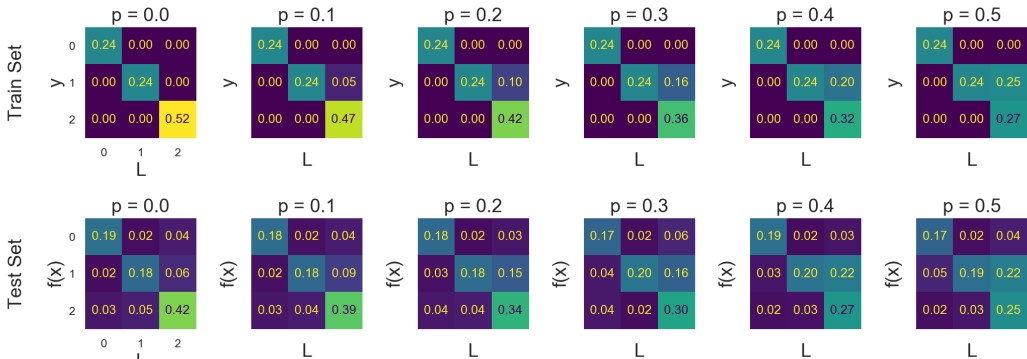

Figure 8: **Feature Calibration for Decision trees on UCI (molecular biology).** We add label noise that takes class 2 to class 1 with probability $p \in [0, 0.5]$. The top row shows the confusion matrix of the true class $L(x)$ vs. the label $y$ on the train set, for varying levels of noise $p$. The bottom row shows the corresponding confusion matrices of the classifier predictions $f(x)$ on the test set, which closely matches the train set, as predicted by Conjecture 1.

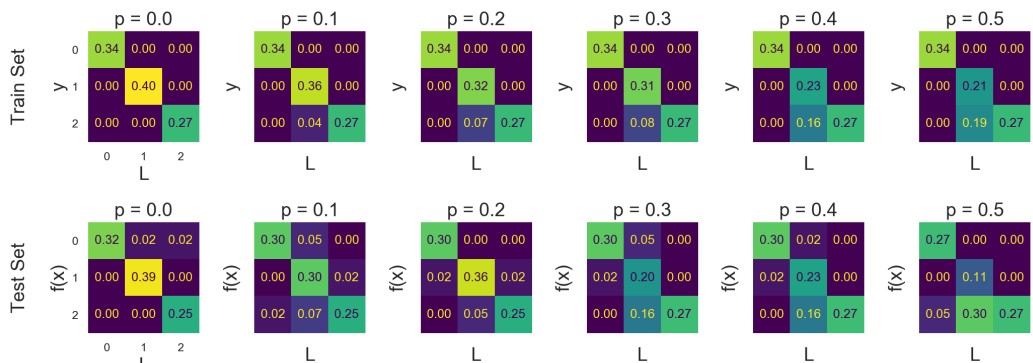

Figure 9: Decision trees on UCI (wine). We add label noise that takes class 1 to class 2 with probability $p \in [0, 0.5]$. Each column shows the test and train confusion matrices for a given $p$. Note that this decision trees achieve high accuracy on this task with no label noise (leftmost column). We plot the empirical joint density of the train set, and not the population joint density of the train distribution, and thus the top row exhibits some statistical error due to small-sample effects.

generalizes well. We initialize the network with a pretrained ResNet-50 from the PyTorch library Paszke et al. (2017) and use the hyperparameters described in Section B.2 to train on this attribute. We then check the train/test joint density with various other attributes like Male, Wearing Lipstick etc. Note that the network is not given any label information for these additional attributes, but is calibrated with respect to them. That is, the network says $\sim 30\%$ of images that have 'heavy makeup' will be classified as 'Attractive', even if the network makes mistakes on which particular inputs it chooses to do so. In this setting, the label distribution is deterministic, and not directly dependent on the distinguishable features, unlike the experiments considered before. Yet, as we see in Figure 11, the classifier outputs are correctly calibrated for each attribute. Loosely, this can be viewed as the network performing 1NN classification in a metric space that is well separated for each of these distinguishable features.

## C.6 COARSE PARTITION

We now consider cases where the original classes do not form a distinguishable partition for the classifier in consideration. That is, the classifier is not powerful enough to obtain low error on the original dataset, but can perform well on a coarser division of the classes.

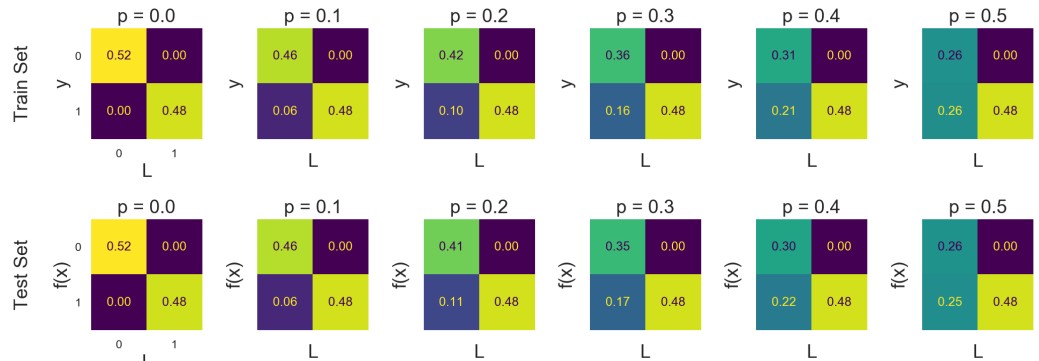

Figure 10: Decision trees on UCI (mushroom). We add label noise that takes class 0 to class 1 with probability $p \in [0, 0.5]$. Each column shows the test and train confusion matrices for a given $p$. Note that this decision trees achieve high accuracy on this task with no label noise (leftmost column).

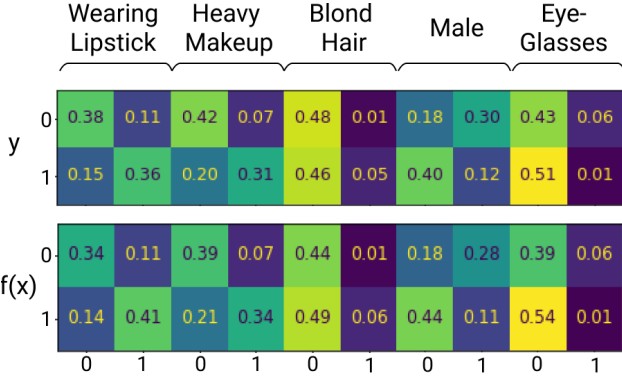

Figure 11: **Feature Calibration for multiple features on CelebA**: We train a ResNet-50 to perform binary classification task on the CelebA dataset. The top row shows the joint distribution of this task label with various other attributes in the dataset. The bottom row shows the same joint distribution for the ResNet-50 outputs on the test set. Note that the network was not given any explicit inputs about these attributes during training.

To verify this, we consider a division of the CIFAR-10 classes into Objects {airplane, automobile, ship, truck} vs Animals {cat, deer, dog, frog}. An MLP trained on this problem has low error ($\sim 8\%$), but the same network performs poorly on the full dataset ($\sim 37\%$ error). Hence, Object vs Animals forms a distinguishable partition with MLPs. In Figure 12a, we show the results of training an MLP on the original CIFAR-10 classes. We see that the network mostly classifies objects as objects and animals as animals, even when it might mislabel a dog for a cat.

We perform a similar experiment for the RBF kernel on Fashion-MNIST, with partition {clothing, shoe, bag}, in Figure 12b.

**ImageNet experiment.** In Table 2 we provide results of the terrier experiment in the body, for various ImageNet classifiers. We use publicly available pretrained ImageNet models from this repository, and use their evaluations on the ImageNet test set.

### C.7  DISCUSSION: PROPER SCORING RULES

Here we distinguish the density-estimation of Conjecture 1 from another setting where density estimation occurs. If $\ell(\widehat{p}, y)$ is a *strictly-proper scoring rule*[3] on predicted distribution $\widehat{p} \in \Delta(\mathcal{Y})$

---

[3]See Gneiting & Raftery (2007) for a survey of proper scoring rules.

| Model | AlexNet | ResNet18 | ResNet50 | BagNet8 | BagNet32 |
|---|---|---|---|---|---|
| ImageNet Accuracy | 0.565 | 0.698 | 0.761 | 0.464 | 0.667 |
| Accuracy on dogs | 0.588 | 0.729 | 0.793 | 0.462 | 0.701 |
| Accuracy on terriers | 0.572 | 0.704 | 0.775 | 0.421 | 0.659 |
| Accuracy for binary {dog/not-dog} | 0.984 | 0.993 | 0.996 | 0.972 | 0.992 |
| Accuracy on {terrier/not-terrier} among dogs | 0.913 | 0.955 | 0.969 | 0.876 | 0.944 |
| Fraction of real-terriers among dogs | 0.224 | 0.224 | 0.224 | 0.224 | 0.224 |
| **Fraction of predicted-terriers among dogs** | 0.209 | 0.222 | 0.229 | 0.192 | 0.215 |

Table 2: ImageNet classifiers are calibrated with respect to dogs: All classifiers predict terrier for roughly $\sim 22\%$ of all dogs (last row), though they may mistake which specific dogs are terriers.

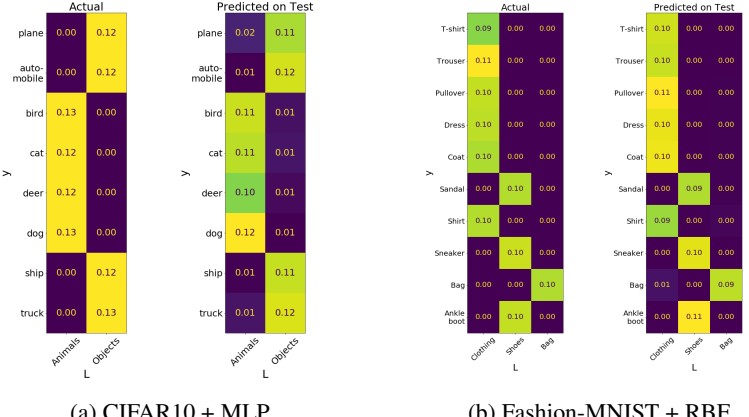

(a) CIFAR10 + MLP                    (b) Fashion-MNIST + RBF

Figure 12: Coarse partitions as distinguishable features: We consider a setting where the original classes are not distinguishable, but a superset of the classes are distinguishable.

and sample $y \in \mathcal{Y}$, then the population minimizer of $\ell(F(x), y)$ is exactly the conditional density $F(x) = p(y|x)$. That is,

$$p(y|x) = \underset{F:\mathcal{X}\to\Delta(\mathcal{Y})}{\operatorname{argmin}} \; \underset{(x,y)\sim p}{\mathbb{E}} [\ell(F(x), y)]$$

This suggests that in the limit of large-capacity network and very large data (to approximate population quantities), training neural nets with cross-entropy loss on samples $(x, y)$ will yield a good density estimate of $p(y|x)$ at the softmax layer. However, this is not what is happening in our experiments. First, our experiments consider the hard-thresholded classifier, i.e. the argmax of the softmax layer. If the softmax layer itself was close to $p(y|x)$, then the classifier itself will be close to $\operatorname{argmax}_y p(y|x)$ – that is, close to the optimal classifier. However, this is not the case (since the classifiers make significant errors). Second, we observe Conjecture 1 even in settings where we train with non-proper scoring rules (e.g. kernel regression, where the classifier does not output a probability).

## D NEAREST-NEIGHBOR PROOFS

### D.1 FEATURE CALIBRATION PROPERTY

*Proof of Theorem 1.* Recall that $L$ being an $(\varepsilon, \mathrm{NN}, \mathcal{D}, n)$-distinguishable partition means that nearest-neighbors works to classify $L(x)$ from $x$:

$$\Pr_{\substack{\{x_i,y_i\}\sim\mathcal{D}^n \\ S=\{(x_i,L(x_i))\} \\ x,y\sim\mathcal{D}}} [\mathrm{NN}_S^{(y)}(x) = L(x)] \geq 1 - \varepsilon \tag{13}$$

Now, we have

$$\{(\mathrm{NN}_S^{(y)}(x), L(x))\}_{\substack{S\sim\mathcal{D}^n \\ x,y\sim\mathcal{D}}} \equiv \{(\widehat{y_i}, L(x))\}_{\substack{S\sim\mathcal{D}^n \\ \widehat{x_i},\widehat{y_i}\leftarrow\mathrm{NN}_S(x) \\ x,y\sim\mathcal{D}}} \tag{14}$$

$$\approx_\varepsilon \{(\widehat{y_i}, L(\widehat{x_i}))\}_{\substack{S\sim\mathcal{D}^n \\ \widehat{x_i},\widehat{y_i}\leftarrow\mathrm{NN}_S(x) \\ x,y\sim\mathcal{D}}} \tag{15}$$

$$\approx_\delta \{(\widehat{y_i}, L(\widehat{x_i}))\}_{\widehat{x_i},\widehat{y_i}\sim\mathcal{D}} \tag{16}$$

Line (15) is by distinguishability, since $\Pr[L(x) \neq L(\widehat{x_i})] \leq \varepsilon$. And Line (16) is by the regularity condition. $\qquad\square$

## E NON-INTERPOLATING CLASSIFIERS: APPENDIX

Here we give an additional example of distributional generalization: in kernel SVM (as opposed to kernel regression, in the main text).

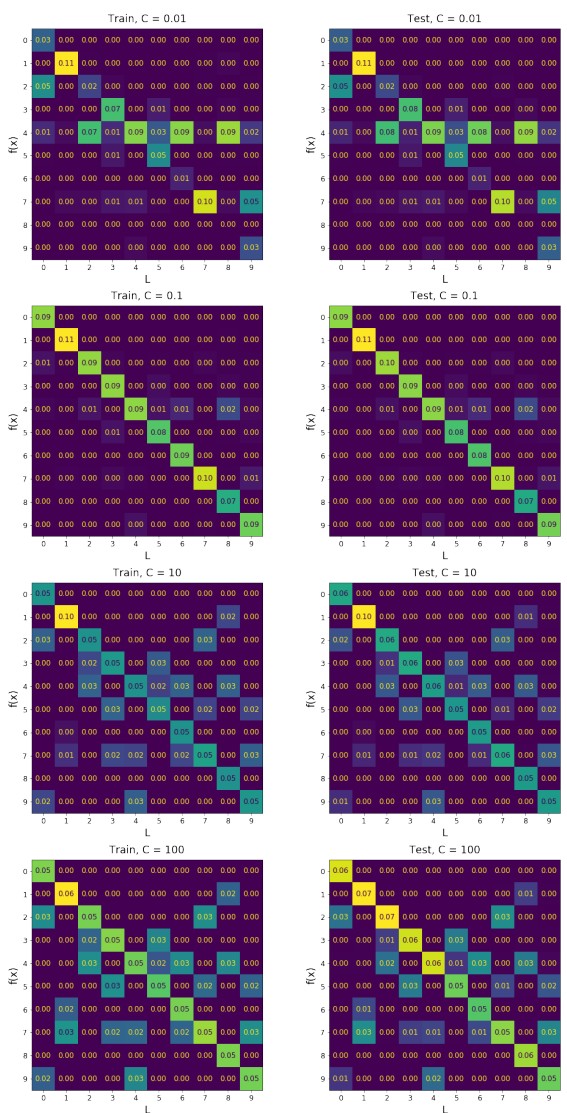

Figure 13: **Distributional Generalization.** Train (left) and test (right) confusion matrices for kernel SVM on MNIST with random sparse label noise. Each row corrosponds to one value of inverse-regularization parameter $C$. All rows are trained on the same (noisy) train set.

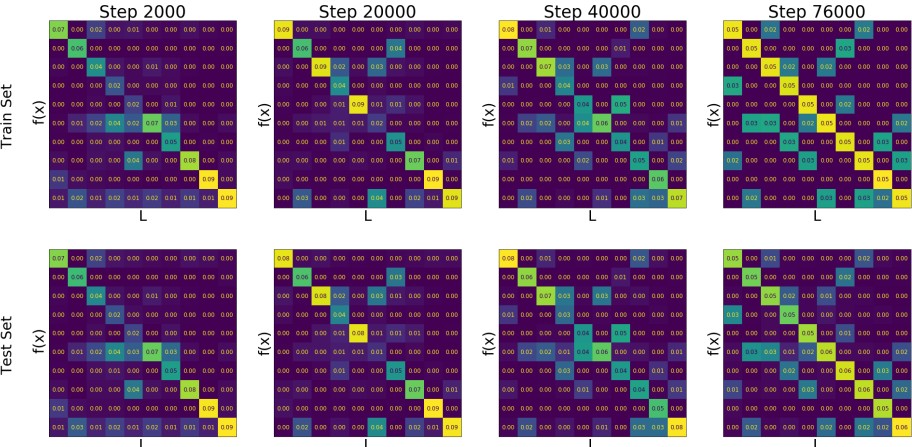

Figure 14: **Distributional Generalization for WideResNet on CIFAR-10.** We apply label noise from a random sparse confusion to the CIFAR-10 train set. We then train a single WideResNet28-10, and measure its predictions on the train and test sets over increasing train time (SGD steps). The top row shows the confusion matrix of predictions $f(x)$ vs true labels $L(x)$ on the train set, and the bottom row shows the corresponding confusion matrix on the test set. As the network is trained for longer, it fits more of the noise on the train set, and this behavior is mirrored almost identically on the test set.

