# OpenReview forum: "Distributional Generalization: Structure Beyond Test Error"
_ICLR.cc/2022/Conference — ICLR 2022 Submitted_

### Official Review · Reviewer_4xds · 2021-11-02

**Correctness:** 4
**Technical Novelty And Significance:** 2
**Empirical Novelty And Significance:** 4
**Recommendation:** 6
**Confidence:** 3

**Main Review:**

Strengths of the paper:
1.	It is well-written and the main ideas are clearly presented. The paper is also quite self-contained.
2.	The proposed empirical observation is novel to the best of my knowledge and a significant observation.
3.	The paper proposes a neatly formalized explanation behind the phenomenon and performs a good set of experiments to back their claim.

Weaknesses:
1.	Based on some efforts to reproduce the results on my end, it is not clear how strongly the proposed observation holds which might limit the significance of the contributions of the paper.

Comments and questions:
1.	What would constitute distributional generalization in the setting of regression? If I consider the setting of regression for a moment, the phenomenon appears to be less surprising: a reasonably smooth regression model which interpolates the train data would necessarily exhibit distributional generalization in that setting.
2.	It would help to see some error bars on the plots in Figure 2B. I tried replicating the toy example (classify CIFAR-10 classes as objects vs animals with label noise on cats) and observed that when label noise is 30% on the train data only 2-5% of cats in test set were being labeled as objects. The network used was a ResNet50 trained to train accuracy 96.2% using SGD with learning rate = 0.1, momentum = 0.9 and weight decay of 5e-4 and trained for ~160 epochs.
However, I did observe that when the label noise is increased to 70%, the distributional generalization effect was seen more strongly (test cats labeled objects 60-80% of the time).
3.	It would make for a stronger case if the paper reports the numbers observed when the label noise experiment is performed on image-net with 1000 classes as well (at least on the non-tail classes). This would further stress test the conjecture. Even if the phenomenon significantly weakens in this setting, the numbers are worth seeing.

Minor comments:
1.	AlexNet top-1 accuracy on ImageNet reported as 56.5%. Isn’t this 63.3%?
2.	Another minor comment is on the name used to describe the phenomenon: distributional generalization sounds a bit strong to capture the empirical phenomenon presented. It represents the ideal of the total variation between the test and train distributions of the network’s outputs vanishing to zero which might not be the case. It is hard to draw this conclusion from a few test functions on which the outputs match.


**Summary Of The Paper:**

The paper reports an interesting phenomenon in deep over-parameterized networks which train to zero train error (known as interpolating networks). Not only does the accuracy on test mimic the accuracy on train, but the accuracy measured on certain subsets of the train set also matches that on the corresponding subsets in test set. The paper coins the term distributional generalization for this phenomenon. The paper then proceeds to perform a number of experiments across a bunch of image datasets to exhibit the generality of this phenomenon.

**Summary Of The Review:**

The empirical phenomenon reported by the paper is novel to the best of my knowledge. It is also an important observation to be made in the behavior of deep neural nets. The paper presents a possible reason behind this behavior and this is formalized in a clean manner. Overall, the paper is very well-written and performs a comprehensive set of experiments to back their claims. I am not entirely convinced the observation holds in the strong form it is stated in but nevertheless I feel the paper’s contributions are sufficient for it to clear the bar for acceptance. Some changes in the empirical experiments which would help make the paper’s case stronger are (i) error bars on plots of label noise experiments (ii) label noise experiments on ImageNet data, (iii) including code in the supplementary material.

---

> ### Author Response · Authors · 2021-11-18
> **Response to Reviewer 4xds**
>
> Thank you very much for your detailed and thoughtful review.
>
> **Re. reproducibility concerns**:
> First, it is important to consider classifiers which interpolate, i.e. reach truly 0% train error.
> Note that 30% label noise on cats is actually only 3% "total noise" on the entire distribution, since cats are only 1/10th of the train set. Your model has ~4% train error (instead of 0%), and this 4% is enough to overshadow the effect size (of 3%). Do you agree?
>
> Second, it is interesting that your results are not symmetric with respect to label noise (e.g. 30% probability of 1 behaves differently than 30% probability of 0). We would expect a properly trained ResNet to behave roughly symmetrically (and indeed, our experiments were nearly symmetric). However, we expect *poorly tuned* models to behave badly, as in this experiment. Just as a sanity check, can you confirm that your exact same training procedure can reach high test accuracy on the original CIFAR-10 problem? (eg 96+% test acc).
>
> Third, note that we have an extensive number of additional experiments in the Appendix, including experiments for kernels and decision trees. Figure 13 (RBF kernel on MNIST) is perhaps the simplest and easiest setting to reproduce, since it is deterministic and fast to train.
>
> Finally, we include all of the experimental details, including hyperparameters in the Appendix. We are also happy to include all of our code in the supplemental (it is standard code).
>
>
>
> **Re. Comments and Questions**:
>
> 1. Stating this in the setting of regression is subtle, and beyond the scope of this paper. However, the natural generalization of DG to regression does not immediately hold – just like Bayes optimality is *incompatibile* with Distributional Generalization for classification, this same statement is true for regression as well. That is, there are settings where the Bayes optimal regressor (i.e. the true regression function) does not satisfy the strongest form of Distributional Generalization.
> 2. (replied to this point above)
> 3. Thank you for this suggestion. Note that we have a number of other experiments in the Appendix, including datasets: CelebA, FashionMNIST, and various UCI tabular tasks. We will consider adding more ImageNet experiments as time allows.
>
>
> **Re. naming**:
> Note that "distributional generalization" is just a definition, and the definition alone does not make claims about when it holds. (Similar to how "classical generalization" is just a definition – some models may satisfy it, some may not). Rather, our "Feature Calibration Conjecture" is what specifies exactly how this definition holds. (Similar to how, say, a "generalization bound" specifies when the definition of generalization is satisfied).

---

> > ### Comment · Reviewer_4xds · 2021-11-22
> > **Thanks for the response**
> >
> > Reproducibility concerns: The classifier I trained achieves over 99% train accuracy and also over 96% test accuracy on the original CIFAR-10 test set. Do you need the train accuracy to be exactly 100% for your results? In any case, there is no harm is adding error bars to your plots (even if the error bar is computed using only 3-4 runs). I understand that resources can be a constraint. I would definitely have liked to have the code to be available along with the submission. That would help alleviate my concerns.
> > Moreover, if you want to claim that this phenomenon is a property of any interpolating model the non-determinism in training ResNets as opposed to RBF kernel on MNIST should not be issue as long as it interpolates.
> >
> > Generalization of DG to Regression: I would be interested to know more about the example where Bayes optimal regressor does not satisfy DG

---

> > > ### Comment · Reviewer_4xds · 2021-11-22
> > > **Follow-up Point**
> > >
> > > In Section 5.2, you seem to suggest that non-interpolating methods might also satisfy DG on certain subsets. This seems to contradict your requirement in your response above that you need the classifiers to be interpolating?

---

> > > > ### Author Response · Authors · 2021-11-25
> > > > **Response2 to Reviewer 4xds**
> > > >
> > > > Thank you very much for engaging with our response.
> > > >
> > > > **Re. reproducibility**: We have uploaded all of our anonymized code here: https://github.com/dg-anon/dg-code-anon
> > > > In particular, the commands for reproducing Experiment 1 are given in the README.md file. Let us know if this addresses your concerns.
> > > >
> > > > **Re. "Example where Bayes optimal regressor does not satisfy DG"**: There are many such examples, but I will describe a particularly simple one below.
> > > >
> > > > Suppose we have the following distribution on (x, y). The marginal on x is standard Normal, in say 100 dimensions (this dim doesn't matter). The conditional p(y|x) is independent of x, and y={+1, -1} randomly, with expectation E[y] = \mu = 0.3.
> > > >
> > > > Note that this is an extremely "easy" regression setting: the regression labels Y are *independent* of the inputs X, and the Bayes-optimal regressor is simply the constant function: $f^*(x) = E[y|x] = \mu = 0.3$.
> > > >
> > > > This constant function does not satisfy distributional generalization, since the distribution of (x, y) is NOT close to (x, f^*(x)). The former distribution is (x, {+1, -1}), while the latter distribution is (x, 0.3). In essence, distributional generalization requires a "sampling-like" property, whereas Bayes-optimality requires a "mean" property. The gap is exactly the difference between sampling from a distribution, and estimating its mean.
> > > >
> > > > This toy example actually reflects what happens in our more realistic experiments. For example, if we apply RBF kernel regression to a label-noised MNIST (regressing to one-hot labels in \R^{10}), the trained model will be closer to a distributional-generalizer than the Bayes-optimal regressor. That is, the trained model will reproduce noisy samples at test-time, rather than "averaging out" the noise as the Bayes would. This can be seen in the bottom-most row of Figure 13, which is the RBF Kernel SVM on noisy-MNIST, with effectively 0 regularization (the experiment yields nearly identical results using the L2 regression loss, instead of the SVM loss, though we didn't include this plot). Let us know if this explanation is unclear.
> > > >
> > > >
> > > > **Re. "Non-interpolating methods may satisfy DG on certain subsets"**: We claim this for "certain" subsets, but the key is, for non-interpolating methods we do not yet understand exactly which subsets. However, for interpolating methods we can predict this precisely (and this is what Conjecture 1 does, via "Distinguishable Features"). This is why we focus on interpolating methods in this paper.

---

### Official Review · Reviewer_SzaL · 2021-11-02

**Correctness:** 3
**Technical Novelty And Significance:** 2
**Empirical Novelty And Significance:** 2
**Recommendation:** 5
**Confidence:** 3

**Main Review:**

It is true that analyzing output distribution of classifier reveals more information than focusing on single metric. The paper provides decent examples to demonstrate the phenomenon where test (output) distribution mirrors that of true labels (supporting conjecture). The paper argues that this result is surprising in contrast to Bayes optimal classifier.
But these results are rather intuitive and not that surprising for "overparametrized" networks. The theoretical analysis of conjecture stops at nearest neighbors and leaves the characterization of classifiers and distributions which obey conjecture to future work. The paper can benefit with more analysis and at least a rough proof sketch for distributional generalization for more general distributions and classifiers. The conjecture is mainly supported by few experiments and not much theory.


**Summary Of The Paper:**

The paper argues about considering entire distribution of classifier behavior rather than a traditional single metric view (test error). The distribution can highlight the areas where the errors actually occur. The paper analyzes datasets with perturbed labels and shows how the classifiers will perform on these perturbed training data. It presents a conjecture that when conditioned on "distinguishable features" the distribution on output is similar to that of distribution of true labels.
The authors present some examples where "distinguishable features" depends on classifier. They prove the conjecture for nearest neighbors classifier under some regularity conditions.

**Summary Of The Review:**

The paper introduces good albeit "unsurprising" conjecture. The paper doesn't support the conjecture with enough theoretical analysis and even the conditions are not fully defined and just relegated to "natural" settings. The paper stands as a good initial study but needs more analysis.

---

> ### Author Response · Authors · 2021-11-18
> **Response to Reviewer SzaL**
>
> Thank you for your review.
>
> Note that reviewers should not call concepts "not surprising" or "intuitive" unless they have been explicitly established before.
> See, for example, this peer review tutorial from ACL: https://aclrollingreview.org/reviewertutorial#6-check-for-lazy-thinking
> *"Many findings seem obvious in retrospect, but this does not mean that the community is already aware of them and can use them as building blocks for future work."*
>
> **Re. "leaves the characterization of classifiers and distributions which obey conjecture to future work."**
> This statement is false: the entire point of our work is to characterize this, and we do so in Conjecture 1.
>
> **Re. "The conjecture is mainly supported by few experiments"**
> We have extensive experiments, including many in the Appendix.
>
> We have models: ResNets, WideResNets, MLPs, RBF Kernels, Laplace Kernels, SVMs, Decision Trees
>
> We have distributions: ImageNet, CelebA, CIFAR-10, FashionMNIST, MNIST, UCI-wine, UCI-mushroom, UCI-molecular-biology
>
> What other experiments do you feel are missing, and what do you hope to learn from them?
>
> **Re. "The conjecture is mainly supported by … not much theory"**
> Our paper should be seen as primarily an empirical paper, meant to inspire later theory.
> There are many examples in the history of science where careful observation and experimentation was the first step towards a deeper theoretical understanding (e.g. Johannes Kepler). Our work is yet another instance of this established research method.
>
> Which is to say: the lack of theory is not a weakness.
>
> We are thus unable to understand the justification for your score. We ask that you increase your score if you agree with our response.

---

> > ### Author Response · Authors · 2021-11-25
> > **Addendum to response**
> >
> > Note, further, that asking for experiments without reason is in violation of ICLR reviewer guidelines: https://iclr.cc/Conferences/2022/ReviewerGuide
> >
> > *Q: Am I allowed to ask for additional experiments?*
> >
> > *A: You can ask for additional experiments, but make sure you justify why they’re necessary to push the paper over the acceptance threshold. New experiments should not significantly change the content of the submission. Rather, they should be limited in scope and serve to more thoroughly validate existing results from the submission.*

---

> > > ### Comment · Reviewer_SzaL · 2021-11-30
> > > **Response to authors**
> > >
> > > Thanks for the response. My main concern in saying "leaves the characterization of classifiers and distributions which obey conjecture to future work" is about section 3.4 where setting is limited to "natural" one.
> > > My thoughts were in line with Reviewer bHpa that in the light of  generalization in learning theory, the conjecture stated in paper is somewhat "expected". I am not claiming to have complete proof for the conjecture.
> > > I do believe paper is interesting and good initial study. With some more characterization of the conjecture and some theoretical analysis (even limited settings), I think it will stand as a good paper. But at current shape, I am sticking with my original rating.

---

### Official Review · Reviewer_bHpa · 2021-11-03

**Correctness:** 3
**Technical Novelty And Significance:** 2
**Empirical Novelty And Significance:** 2
**Recommendation:** 3
**Confidence:** 3

**Details Of Ethics Concerns:**

I have a concern regarding anonymization. In page 4, the authors say "Since the appearance of the current work on arXiv, it has been directly built on by other authors. The work of Jiang et al. (2021) investigates our conjectures further, and extends them to develop a method for out-of-distribution uncertainty estimation. This highlights the fundamental nature and importance of our results, since they have already been used in a practical application".

The paper is highlighting this follow-up to the extent that it seems like a reviewer should look at the follow-up to evaluate the significance of the paper itself, but glancing at the mentioned Jiang et al. (2021) work would probably directly reveal the identity of the authors of this paper. I'm personally ignoring this discussion for my evaluation to preserve anonymity, but I'm not sure of the right policy here, and whether this discussion was appropriate.

**Main Review:**

The feature-calibration conjecture seems interesting at first sight, but I'm not sure it is really saying that much, which is my main concern for the paper.

Though generalization in learning theory is often summarized as just train performance $\approx$ test performance, what it really talks about is concentration of measure. With this in mind, if we define any subgroup of the data (such as in Experiment 1, where the subgroups are the different animals), then if the model is generalizing it's training and test performance on the subgroups will be approximately equal---I'm not sure why the authors are regarding this experimental finding as surprising or novel. This just follows from the fact that the generalization bounds hold for all subsets in any partition of the space as well, at most suffering some union bound to get uniform bounds for all subgroups in the partition. The authors don't repeat Experiment 1 for the MLP model which is not expressive enough to identify the different sub-groups, but they repeat a similar experiment in Fig. 6. For the MLP model, the training and test numbers differ by at most 1-2%, as one would expect based on generalization bounds. If one argues that the numbers differ slightly less across the groups in Fig. 5, that seems just a consequence of the fact that the model in Fig. 5 has an overall generalization gap which is very small (smaller than a percent it seems), and therefore even if one does a union bound across the 10 subgroups we will still get very small error on all groups. One gets a tiny bit more in Fig. 6 but I'm not sure if that is so important. And if the authors believe somehow that this difference is in fact important, then I think the entire argument needs to be rewritten to take this into account.

To summarize, an interpolating model will match the training distribution, and if it is generalizing, it will match the test distribution on any not too large partition of the space (to avoid too large factors in the union bound). Therefore, Fig. 1 is not surprising, given the fact that models which interpolate can still generalize. Of course, there is the mystery of why interpolating models generalize at all, but that's another story.

There are strengths to the paper too, the definition and subgroup property is interesting, and the experiments are well carried out (though it seems that the authors don't include any code). There could be merit in exploring the idea further, but at present it falls short of the mark in my opinion.

**Summary Of The Paper:**

The paper proposes a new notion of ``distributional generalization". It formalizes this through a conjecture (feature calibration conjecture) which says that the output distribution of an interpolating classifier matches the distributions of the labels on a certain class of sub-groups of the data. The paper evaluates this conjecture empirically, and also proves it for certain nearest neighbor models.

**Summary Of The Review:**

As mentioned below, the overall framework is not convincing to me and hence I recommend rejection.

---

> ### Author Response · Authors · 2021-11-18
> **Response to Reviewer bHpa**
>
> Respectfully, **the reviewer has made bold claims about relations to prior work, with absolutely no substantiating evidence or references.**
> If the reviewer believes our results "just follows from … generalization bounds", we encourage the reviewer to produce either:
> 1. A proof of our Feature Calibration Conjecture.
> 2. Any reference to prior work which implies the Feature Calibration Conjecture.
>
> If the reviewer does this, we are happy to add them as a co-author for their significant contributions to statistical learning theory.
> In absence of this, the reviewer has no basis for claiming our results follow from standard facts.
> Such claims must be substantiated with evidence.
>
> Finally, it is impossible to show that Distributional Generalization follows from classical generalization (as the reviewer suggests).
> This is because these two concepts are *inconsistent* with each other: in Figure 2A, for example, the trained model has very high generalization gap (since it interpolates its train set), but still satisfies distributional generalization. Does this make sense?

---

> > ### Comment · Reviewer_bHpa · 2021-11-29
> > **Response to comment**
> >
> > I would have appreciated if the authors tried to understand my comments and addressed specific points.
> >
> > I'll repeat what I said in my review, maybe this stems from some misunderstanding and a rephrasing will help. Define any partition of the data domain into some collection of sets. The generalization guarantees for any algorithm hold over any given set in this partition---in the same way which they hold over the overall data distribution. By suffering a union bound over all sets in the partition, the bound holds for all sets in the partition too. Repeating what I said earlier, in Fig. 6. in the appendix, for the MLP model, the training and test numbers differ by at most 1-2%, as one would expect based on generalization bounds. If one argues that the numbers differ slightly less across the groups in Fig. 5, that seems just a consequence of the fact that the model in Fig. 5 has an overall generalization gap which is very small (smaller than a percent it seems), and therefore even if one does a union bound across the 10 subgroups we will still get very small error on all groups. One gets a tiny bit more in Fig. 6 but I'm not sure if that is so important. As said earlier, it is possible to maybe make a point that that this difference is in fact important, but the authors don't seem to think that this is the point.
> >
> > I don't get the point about 2A made in the comment. Interpolation is not inconsistent with generalization. Interpolating models such as neural networks do in fact get a very small generalization gap, why they do is a separate mystery.

---

> > > ### Author Response · Authors · 2021-11-30
> > > **Response2 to Reviewer bHpa**
> > >
> > > Thank you for rephrasing, I think I grasp your misunderstanding now. Let us know if the following explanation clarifies it.
> > >
> > > Here is the point: In Figure 6, there is actually a *very large* generalization gap. The MLP fits its train set perfectly (reaching train error 0%), and yet it has very "noisy" test outputs (and has test error ~37%). Thus, it is not in fact true that this MLP "generalizes well" in the classical sense – the train error is very far from the test error.
> > > However, the MLP generalizes well in a "distributional" sense (within 1-2%, as you say), and this is the point of our paper.
> > > Our paper formalizes exactly the observation that you made, which is that the top and bottom matrices in Figure 6 are close to each other. However, these matrices are NOT depicting the train/test error. Rather, they show the *joint distributions* of (x, f(x)) on the train set, vs. on the test set.
> > >
> > > To recap, the point of our paper is to argue that in Figure 6 (as well as in Figure 1 and in Figure 5), classifiers actually have very *large* generalization gaps, and yet have "good behavior" in a certain distributional sense.
> > >
> > >
> > > Re "Interpolating models such as neural networks do in fact get a very small generalization gap">
> > > This is not universally true. Here is an extreme example, similar to the ones in our paper, to illustrate the point. Consider the famous [Zhang et al. 2016] experiment: train a large network to interpolate a *randomly labeled* train set. This network achieves 0% train error, but trivial test error (since it will behave randomly at test-time). Thus, this interpolating network has a very large "classical generalization gap."
> > >
> > > On the other hand, this network satisfies "distributional generalization": Distributional generalization says that a classifier trained on randomly-labeled inputs (at train time) should produce randomly-labeled outputs (at test time). The train and test *distributions* (as we define in our paper) are close, even though the difference between train and test *error* is large.
> > >
> > > To summarize, the point of these examples is that, when we shift perspective from considering just *error* to considering entire *distributions*, our conclusions about when "generalization" occurs can change.

---

> > > > ### Comment · Reviewer_bHpa · 2021-11-30
> > > > **Response**
> > > >
> > > > Thanks for the response, but I don't see how I have misunderstood. On any given subset of the data, the training loss and test loss would be similar for a model which generalizes (where generalization means behaving similarly on training and test, certainly if there's noise added in train but not in test the behavior would be different). Therefore, on subsets where noise is not added, the accuracy would remain the same. To emphasize, this does follow directly from known generalization bounds, for e.g. for any model where VC theory is sufficient to explain generalization it would directly give generalization bounds on any subset of the data.
> > > >
> > > > It's correct that it's surprising that interpolating models generalize given the experiments in Zhang'16, but as I said before that is an orthogonal story to the paper, and is not the point being made here---given that the neural networks can generalize despite having the capacity to overfit, generalization on subsets is expected.

---

> > > > > ### Author Response · Authors · 2021-12-01
> > > > > **Reply**
> > > > >
> > > > > There is a fundamental reason that generalization bounds cannot imply the behavior we see in, for example, Figure 6 (and more generally, the entire paper).
> > > > >
> > > > > As already described, the reason is: The experiments in our paper give cases where classifiers **do not generalize** in the classical sense. There is a large gap between train error and test error (~37% in Figure 6). Thus, no bound on "generalization gap" can imply this behavior, simply because our models **do not** generalize well in the classical sense.
> > > > >
> > > > > However, they do have certain other "good behavior", and the point of our paper is to identify and formalize this behavior.

---

> > > > ### Comment · Reviewer_4xds · 2021-12-01
> > > > **Doesn't distributional generalization imply a small generalization gap?**
> > > >
> > > > Doesn't the fact that the model in Figure 6 exhibits distributional generalization on each of the input classes imply that it should have a small generalization gap as well?

---

> > > > > ### Author Response · Authors · 2021-12-01
> > > > > **Reply**
> > > > >
> > > > > No, it does not.
> > > > > Perfect Distributional Generalization (for interpolating models) is statement that the following two distributions are equivalent $(x, f(x)) \equiv (x, y)$.
> > > > >
> > > > > Whereas the Bayes-optimal classifier would satisfy: $(x,f(x)) \equiv (x, y^*(x))$
> > > > > Where $y^*(x) = \textrm{argmax}_y p(y|x)$ is the Bayes-optimal decision for x.
> > > > >
> > > > > These two statements are *not* equivalent in any setting with non-zero Bayes risk (for example, with label noise in our experiments).
> > > > >
> > > > > To be explicit: Interpolating models with high test error will necessarily have high "classical generalization" gap (since they have train error 0%, but high test error). However, it is still possible to satisfy distributional generalization, if the classifier outputs f(x) approximate a *sample* from $p(y|x)$. This is exactly what's happening in Figure 6.
> > > > >
> > > > > The difference between classical generalization and DG is, in some sense, the difference between "classification" and "sampling."

---

> > > > > > ### Comment · Reviewer_4xds · 2021-12-02
> > > > > > **Reply**
> > > > > >
> > > > > > Thanks for the response!
> > > > > > I'm not sure if Figure 6 is the best example to demonstrate this point. It is not fair to show CIFAR-10 test error as an example of high generalization gap in the context of Figure 6 as the classification problem on which DG is being evaluated on only has two classes (as opposed to 10 classes in calculation of the CIFAR-10 test error). And from the table in Figure 6, it seems like the test error for this binary problem would only be around 12%.
> > > > > > But the random labels experiment of Zhang et. al. 2016, does demonstrate a setting where DG holds but generalization gap is large and perhaps better illustrates this point.

---

### Decision · Program_Chairs · 2022-01-20

**Decision:**

Reject

**Comment:**

The paper proposes a new perspective on the generalization performance of interpolating classifiers based on the entire joint distribution of their inputs and outputs. It conjectures that, when conditioned on certain subgroups, the output distribution matches the distribution of true labels. The conjecture is investigated empirically on a number of datasets and models, and proved to hold for a simple nearest neighbor model.

This paper generated varying responses from the reviewers and a detailed discussion. One main concern focused on whether the feature calibration conjecture is actually surprising, given standard expectations about generalization from learning theory. Indeed, from the discussion and the paper itself, it seems the authors conceived of classical generalization as a statement about whether train performance $\approx$ test performance, whereas one reviewer remarked that "what it really talks about is concentration of measure." I agree with the importance of this distinction in general, though it is perhaps less relevant in the current setting of modern interpolating classifiers, for which so little about generalization is understood in the first place. In particular, the empirical observations of varied forms of good generalization behavior for overparameterized models are likely to be interesting to the community, regardless of whether this behavior might be expected in the large sample limit.

As such, this is a very borderline paper, with many good arguments both for and against acceptance. After a detailed discussion among the chairs, it was decided that the current version is just shy of the acceptance threshold, but I would strongly encourage the authors to address the main reviewer concerns and resubmit a revised manuscript to a future venue.